# Dpp and Hedgehog promote the glial response to neuronal apoptosis in the developing *Drosophila* visual system

**Sergio B. Velarde**, **Alvaro Quevedo**, **Carlos Estella**, **Antonio Baonza** *

Centro de Biología Molecular Severo Ochoa, Consejo Superior de Investigaciones Cientificas (CSIC)/Universidad Autonoma de Madrid (UAM), Madrid, Spain

* abaonza@cbm.csic.es

**Data Availability Statement:** All relevant data are within the paper and its Supporting Information files.

## Abstract

Damage in the nervous system induces a stereotypical response that is mediated by glial cells. Here, we use the eye disc of *Drosophila melanogaster* as a model to explore the mechanisms involved in promoting glial cell response after neuronal cell death induction. We demonstrate that these cells rapidly respond to neuronal apoptosis by increasing in number and undergoing morphological changes, which will ultimately grant them phagocytic abilities. We found that this glial response is controlled by the activity of Decapentaplegic (Dpp) and Hedgehog (Hh) signalling pathways. These pathways are activated after cell death induction, and their functions are necessary to induce glial cell proliferation and migration to the eye discs. The latter of these 2 processes depend on the function of the c-Jun N-terminal kinase (JNK) pathway, which is activated by Dpp signalling. We also present evidence that a similar mechanism controls glial response upon apoptosis induction in the leg discs, suggesting that our results uncover a mechanism that might be involved in controlling glial cells response to neuronal cell death in different regions of the peripheral nervous system (PNS).

## Introduction

A complex nervous system is comprised of neurons and glial cells whose development and function are mutually interdependent. The intricate interaction between these 2 cell types is essential for the generation and maintenance of a functional nervous system. During development or after neuronal damage, cells within the nervous system undergo changes in order to preserve structural integrity and function. Glial cells actively participate in all aspects of nervous system development, including mechanisms involved in maintaining structural robustness and functional plasticity. In response to neuronal damage, glial cells proliferate, change their morphology, and alter their behaviour [1–4]. This glial cell response is associated with their regenerative function and is found across different species. The signalling pathways underlying glial response and how they are coordinated remain poorly understood. We can have a better understanding of the mechanisms involved in regulating this process by exploring how glial cells respond to the induction of neuronal apoptosis.

**Funding:** This study was supported by grants from: Fundación Ramón Areces (to AB); Programa Estatal de Generación de conocimiento y fortalecimiento científico y tecnológico del sistema de I+D+I (Ministerio de Ciencia, Innovación y Universidades) grants PGC2018-095144-B-I00 (to CE) and BFU2014-54153-P (to AB); The National Council of Science and Technology from México (CONACyT) (to SBV); Fellowship from the Ignacio Larramendi foundation (to SBV); Consejeria de Ciencia. Universidades e innovacion, Comunidad Autonoma de Madrid. "Ayudas para la contratacion de Investigadores Predoctorales" (to AQ). The funders had no role in study design, data collection and analysis, decision to publish, or preparation of the manuscript.

**Competing interests:** The authors have declared that no competing interest exist.

**Abbreviations:** AEL, after egg laying; CNS, central nervous system; Dpp, Decapentaplegic; EdU, 5-Ethynyl-2-deoxyuridine; GMR, glass multiple reporter; GRR, glial regenerative response; *hep*, *hemipterous*; Hh, Hedgehog; JNK, c-Jun N-terminal kinase; LC3, light chain 3; MF, morphogenetic furrow; PN, perineurial; PNS, peripheral nervous system; *ptc, patched*; *puc, puckered*; RNAi, RNA interference; *rpr, reaper*; SEM, standard error of the mean; TNF, tumour necrosis factor; WG, wrapping glia.

*Drosophila melanogaster* is an excellent model system to discover evolutionarily conserved gene functions and gene networks. Previously, the eye disc of *Drosophila* has been used to analyse basic mechanisms regulating the migration of glial cells along their neuronal partners [2,5–11]. The eye disc develops from a group of ectodermal cells with embryonic origin, distinct from the neuroectodermal cells that form the central nervous system (CNS). Thus, unlike its mammalian counterpart, *Drosophila* eyes are not part of the CNS. Nevertheless, similarly to mammalian systems, they do contain neurons and glial cells. The eye primordia develops progressively, from posterior to anterior, over the course of approximately 2 days. A morphogenetic furrow (MF) sweeps across the disc during this period, leaving in its wake, developing clusters of photoreceptor cells that will become the individual units of the compound eye, known as ommatidia [12]. Therefore, while the region anterior to the furrow is mainly composed of proliferating, undifferentiated cells, the region posterior to the furrow consists predominantly of cells that have exited the cell cycle and have begun to differentiate into photoreceptors [12–14]. Unlike photoreceptors, the progenitors of all subretinal glia are not generated from the eye disc cells. During early embryonic stages, the anlage of the eye disc is established, and a few glial cells are born in the initial segment of the Bolwing nerve, which will later become the optic stalk and serve to connect the developing imaginal disc to the brain [9,10,15]. These new glial cells are the precursors of the eye disc glial cells. During larval stages, these precursor cells proliferate, forming new glial cells that accumulate in the optic stalk. As the eye imaginal disc grows and neurogenesis is initiated behind the MF, glial cells leave the optic stalk and migrate onto the eye disc [9,10,11,15].

The eye discs contain distinct glial cell types [10]. Subperineurial cells, the so-called carpet cells, are 2 large cells that cover the entire differentiated part of the eye disc epithelium. Sitting basally to these 2 cells are the perineurial (PN) glial cells that have a distinct morphology in the optic stalk and in the eye disc [10,16]. These cells define a reserve pool, which can generate glial cells when necessary (for plasticity and development); accordingly, these cells maintain the ability to divide during eye disc development [10]. In addition, carpet cells separate the PN glia from the underlying wrapping glia (WG), a group of cells that derive from the PN glial and enwrap all axons produced by the photoreceptors. The WG cells perform functions that resemble the non-myelinating Schwann cells forming Remak fibers in the mammalian peripheral nervous system (PNS) [7]. Likewise, Schwann cells play a key role in promoting regeneration and provide the high ability of the peripheral nerves to regenerate [17].

Many studies using eye discs as a model system have contributed to uncovering the basic mechanisms and signalling pathways involved in the coupling of neuronal and glial development and have shown that *Drosophila* glial cells can serve as an experimental model to gain insights into mammalian glial biology [6,7,10,18,19]. However, despite the widespread use of this model, little is known about the response of glial cells upon apoptosis induction of neural tissue and the signalling pathways that might be mediating this function. The unique developmental features of the fly eye disc make this structure an excellent model to analyse the signals emitted by dying neural tissue and the mechanisms involved in regulating glial cell response. Furthermore, considering the similarities between non-myelinating Schwann cells and WG, the eye discs might provide new insights into the signalling pathway network involved in regulating glial cells behaviour in response to apoptosis induction in the PNS.

Here, we use the eye disc to explore the mechanisms involved in promoting glial cell response to the induction of neural apoptosis during development. We demonstrate that eye glial cells respond by increasing in number and undergoing morphological changes that confer them phagocytic activity. We found that this glial response is controlled by the activity of the Decapentaplegic (Dpp) and Hedgehog (Hh) signalling pathways. These pathways are activated in glial cells upon apoptosis induction in the retina region, and their function is necessary for

stimulating the proliferation and migration of glial cells to the eye discs. This latter process depends on the function of the c-Jun N-terminal kinase (JNK) pathway, which is activated by Dpp signalling. Remarkably, we present evidence indicating that a similar mechanism controls glial response upon apoptosis induction in the leg discs. As in the eye discs, most leg glial cells are born in the CNS/PNS transition zone during larval stages and migrate into the forming leg [16]. We observed that after apoptosis induction in leg discs, glial cells accumulate in this region. The function of Dpp and JNK, but not Hh signalling, is required for this glial response; hence, our results uncover a mechanism that might be involved in controlling glial cell response to neural cell death in different regions of the PNS.

## Results

### Cell death induction in the eye disc epithelium promotes the accumulation of glial cells

Different studies have shown that when apoptosis is induced in the retinal region of eye discs, a regenerative response that includes compensatory proliferation is initiated [20]. However, whether this response also involves glial cells activation was unknown. In order to study glial cell response to apoptotic induction in this model system, we have examined the localisation, pattern of proliferation, and number of glial cells in eye discs after inducing apoptosis in the retinal region. To genetically induce targeted cell death in this region of the eye disc epithelium, we used the *Gal4/UAS/Gal80^{ts}* system to transiently overexpress the proapoptotic gene *reaper* (*rpr*) under the control of the eye specific glass multiple reporter (GMR) *Gal4* line. *GMR-Gal4* is expressed in all cells posterior to the MF, including photoreceptor cells (Fig 1A and 1A"). Given that glial cells do not originate in the eye disc, this driver is not active in glial cells (Fig 1A'). Therefore, cells in the retinal region would nonautonomously induce changes in the behaviour of glial cells. We used the *tub-Gal80^{ts}* transgene to modulate the time of apoptotic induction to 72 hours before dissection (see Materials and methods). As expected, we found that in *GMR-Gal4 tub-Gal80^{ts} UAS-rpr*, larvae cell death strongly increased behind the MF (S1 Fig). Remarkably, we observed a pronounced increase in the density of glial cells in damaged discs compared to control discs ($0.019 \pm 0.0004$, $n = 47$ versus $0.009 \pm 0.0003$, $n = 39$ number of glial cells/area $\mu^2$ occupied by glial cells, in damaged discs and control discs, respectively, $p < 0.0001$; Fig 1D and 1G, Table A in S1 Text). In contrast to control discs, where glial cells are always located on the basal layer of the discs (Fig 1C–1C""), injured discs contained glial cells contacting with photoreceptors in the middle layer of the eye disc epithelium (yellow arrowheads in Fig 1D"–1D""), as well as in the apical region (green arrowheads in Fig 1D"–1D""). Interestingly, retinal damage also affects the position of glial cells with respect to the MF. Thus, in control discs, the anterior border of glial migration lies between 0 and 5 cell rows behind the most anterior row of photoreceptors, whereas in damaged discs, the anterior boundary of the glial migration moved ahead (Fig 1, compare 1E–1E' with 1F–1F' and 1H).

In order to get a better understanding of the sequence of events associated with glial response, we examined glial behaviour at different times after induction of apoptosis in the retinal region. To that end, *GMR-Gal4 tub-Gal80^{ts} UAS-rpr* larvae were raised at permissive temperature (17°C), then shifted to the restrictive temperature (29°C) during 24 hours to induce *UAS-rpr* and then shifted back to the permissive temperature to allow time for recovery (see Materials and methods). Discs were analysed at different time points: immediately after *rpr* induction (T0), after 24 hours at permissive temperature (T1), and after 48 hours of recovery (T2) (Fig 2M). In discs dissected immediately after genetic ablation (T0), we observed a large number of apoptotic cells and cellular debris labelled with anti-Dcp1 (Fig 2C–2D"). Most apoptotic cells were located at the basal region of the discs and predominantly corresponded

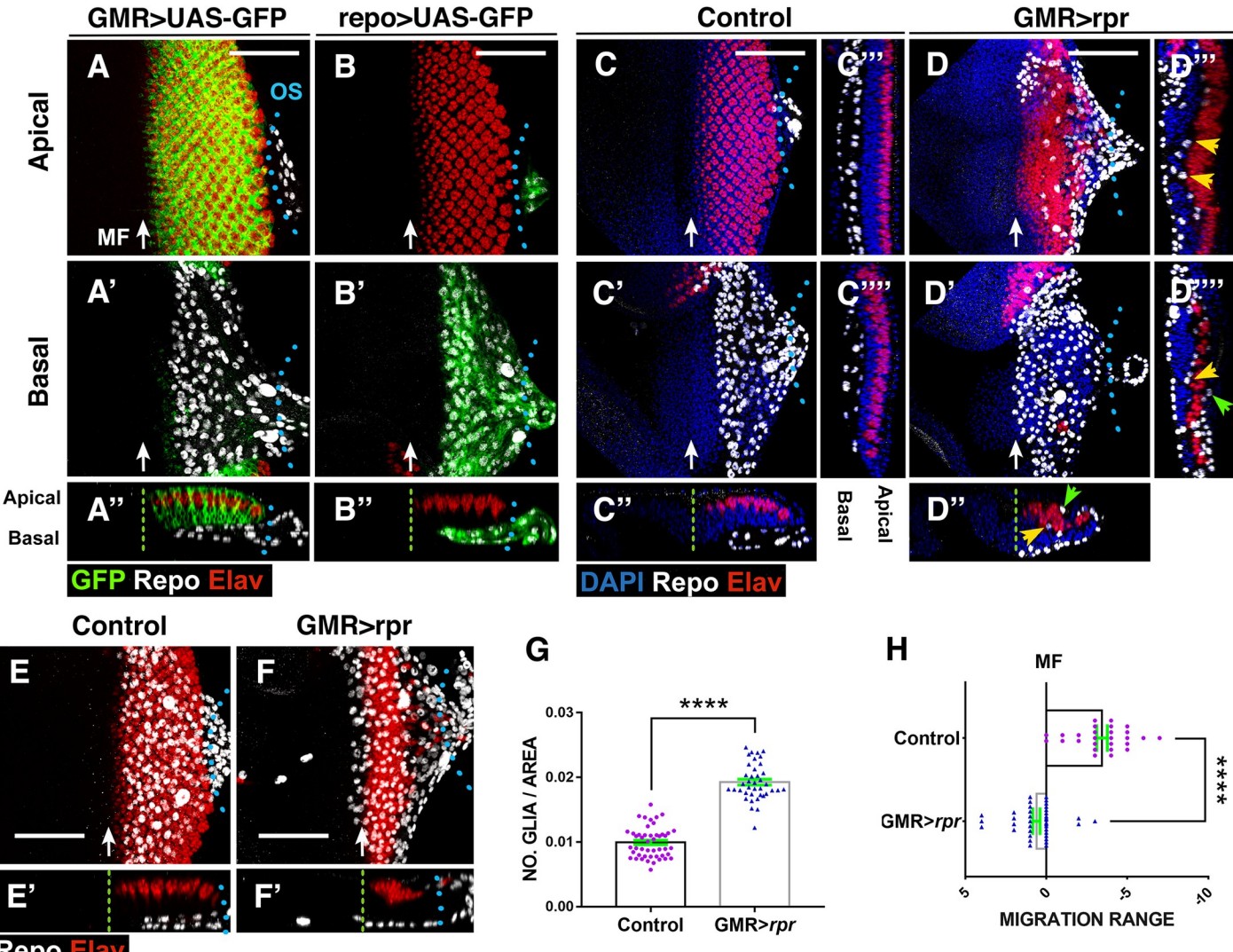

**Fig 1. Response of glia to neuronal apoptotic induction in the eye discs.** (A–F') Third instar eye discs stained with anti-Elav (Photoreceptors in red), anti-Repo (glial cells in white), and the nuclear marker DAPI (blue). (A–D) Apical/middle layers of the eye disc epithelium. (A'–D') Basal layers of the eye discs. (A"–D", E'–F') The X–Z projections show cross sections perpendicular to the furrow of the eye discs of panels A–D (A"–D") and E–F (E'–F'). (C'"–C"" and D'"–D"") Transverse sections parallel to the furrow of discs shown in C (C'"–C"") and D (D'"–D""). (A–A") Eye disc showing the expression of *UAS-GFP* (green) under the control of *GMR-Gal4*. Glial nuclei (anti-Repo in white) are located in the basal layer of the eye disc, and they do not express *UAS-GFP*. (B–B") The expression of *UAS-GFP* (green) under the control of *repo-Gal4* is restricted to glial cells. (C–C"") In control discs, subretinal glial cells are always located in the basal layer of the disc. (D–D"") In damaged *UAS-rpr/+ GMR-Gal4, tub-Gal80^{ts}* eye discs glial cells are not only in the basal layer of the disc but also in the middle and apical layers. This is most clearly seen in transverse sections shown in (D"–D""). Arrowheads indicate glial cells located in apical (green arrowheads) and middle (yellow arrowheads) layers of the discs (D"–D""). Note that glial cells in the medial layer directly contact photoreceptors. (E–F') Projections of confocal images of the basal layers of third instar control (*GMR-Gal4 tub-Gal80^{ts}*) (E) and damaged (*UAS-rpr/+ GMR-Gal4, tub-Gal80^{ts}*) (F) eye showing the relative position of the anterior border of glial migration with respect to the anterior most row of photoreceptors. In damaged discs, the anterior border of glial migration lays 2–4 row in front of the MF (green dashed line). (G) Graph shows the density of glial cells (number of glial cells/area occupied by glial cells in μm²) in control and damaged *UAS-rpr/+ GMR-Gal4 tub-Gal80^{ts}* /+ discs. (H) Graph shows the relative position of the anterior border of glial migration with respect to the anterior most row of photoreceptors (0 indicates the position of this row) in control and damaged *UAS-rpr/+ GMR-Gal4 tub-Gal80^{ts}* eye disc. Error bars represent SEM. Statistical analysis is shown in Table A in S1 Text. In this and all subsequent figures, anterior is to the left. Blue dotted lines indicate the approximate limit between eye disc and optical stalk (os). The green dashed line indicates the approximate position of the MF. Scale bars, 50 μm. The numerical data used in this figure are included in S1 Data. GMR, glass multiple reporter; MF, morphogenetic furrow; *rpr*, *reaper*; SEM, standard error of the mean.

to interommatidial cells, as they did not express the neuronal marker Elav (Fig 2C–2D"). The density of glial cells in these discs was significantly higher than in control discs (0.01 ± 0.00033 number of glial cells/area μ² in control discs, *n* = 28 versus 0.015 ± 0.0009 in damaged discs

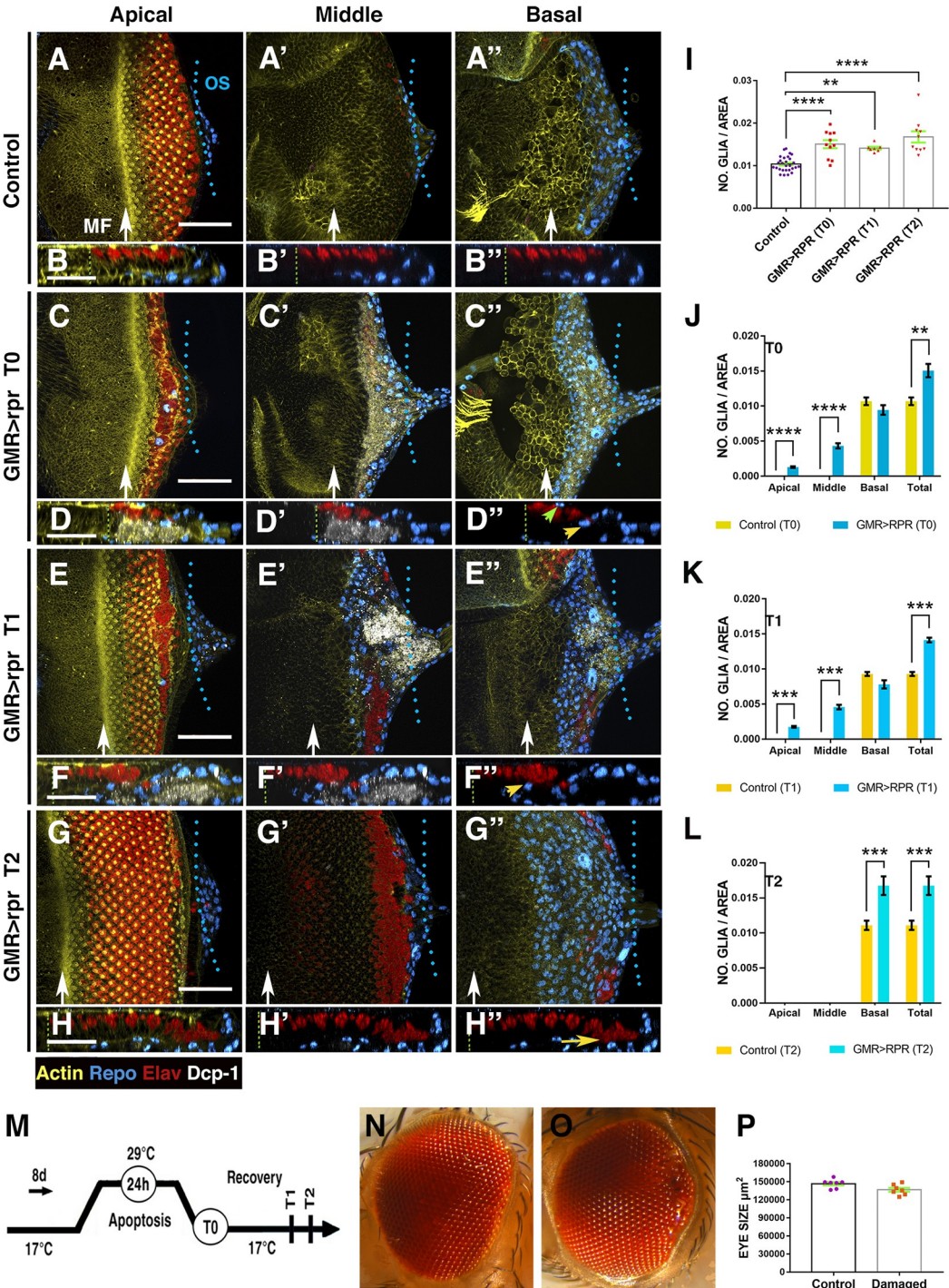

**Fig 2. Overview of glial response at different times after apoptotic induction in the eye discs.** (A–H") Third instar eye discs stained with anti-Elav (red) and anti-Repo (blue), the apoptotic marker Dcp-1 (grey), and Phalloidin to visualise F-actina (yellow). Control undamaged disc (*GMR-Gal4 tub-Gal80^ts*) (A–B") and *UAS-rpr/+; GMR-Gal4 tub-Gal80^ts/+* eye discs analysed at different times after inducing apoptosis (C–H"). (A, C, E, and G) Apical, (A', C', E', and G') middle, and (A", C", E", and G") basal layers of the eye disc epithelium. (B–B", D–D", F–F", and H–H") X–Z projections show a cross section of the eye discs epithelium perpendicular to the furrow of the discs shown in A (B–B"), C (D–D"), E (F–F"), and G (H–H"). (I) Bar charts show the average density of glial cells of discs analysed at different times after apoptotic induction. (J–L) Bar charts show the average density of glial cells and their apical/basal localisation of discs immediately after apoptotic induction T0 (J), discs analysed after 24 hours of recovering T1 (K), and discs examined after 48 hours of recovering T2 (L). In each graph is also shown the total density of glial cells for control (*GMR-Gal4 tub-Gal80^ts/+*) and *UAS-rpr/+; GMR-Gal4*

*tub-Gal80$^{ts}$/+* eye discs. (M) Schematic diagram of the temperature shifts used in this experiment. (A–B'") In control discs, subretinal glial cells are always localised in the basal layer of the eye disc. (C–D") *UAS-rpr/+; GMR-Gal4 tub-Gal80$^{ts}$* eye discs dissected and analysed immediately after ablation (T0). Apoptotic cells are located in the middle and basal layer of the discs, whereas glial cells appear in the middle (yellow arrowhead in D") as well as apical planes (green arrowhead in D") (D–D"). (E–F") *UAS-rpr/+; GMR-Gal4 tub-Gal80$^{ts}$/+* eye discs analysed after 24 hours of recovering. The new rows of photoreceptors that are specified during the recovery time are not affected. Note that most cellular debris was displaced towards the posterior region of the discs and even inside the optic stalk. The average density of glial cells in these discs is still higher than in control discs (K). We find a high number of glial cells in apical and middle layers of the discs (yellow arrowhead in F"). (G–H") Damaged eye discs analysed after 48 hours of recovering. In these discs, we do not find apoptotic debris (in grey) (H–H"). In the posterior region of the disc, we observed very disorganised photoreceptors, which are located in the basal layer and that likely correspond to the photoreceptors that did not die after overexpressing *UAS-rpr* (yellow arrow in H"). (N–O) Adult control eye (N) and an eye derived from damaged eye discs (O). (P) Bar charts show the average size of control eyes and adult eyes developed from damaged discs. Statistical analysis is shown in Table B in S1 Text. In this and all subsequent figures, white arrows indicate the approximate position of the MF. Scale bars, 50 μm. The numerical data used in this figure are included in S1 Data. GMR, glass multiple reporter; MF, morphogenetic furrow; *rpr*, *reaper*.

*n* = 11, *p* = <0.0001; Fig 2C–2D", 2I, and 2J). This is mainly due to the increased number of glial cells that appear in the middle/apical layers of the disc epithelium (Fig 2J). After 24 hours of recovery at permissive temperature (T1; Fig 2E–2F"), the density of glial cells was still higher in damaged discs than in control discs; however, numbers were similar to those found at T0 (0.015 ± 0.0009 number glial cells/area μ$^2$ at T0 versus 0.014 ± 0.00033 number glial cells/area μ$^2$ at T1; Fig 2I and 2K). We also observed a high number of glial cells in the middle/apical layers of the retinal epithelium (Fig 2E–2F"). After 48 hours of recovery (T2), we did not see apoptotic cells or cellular debris in the epithelium (Fig 2G–2H"). The density of glial cells was similar to the earlier time points, but higher when compared to control discs (0.01 ± 0.00033 number glial cells/area μ$^2$ in control discs versus 0.016 ± 0.0013 number glial cells/area μ$^2$ at T2; Fig 2G–2H", 2I and 2L). In contrast to the discs analysed at T0 and T1, we did not find glial cells in the middle or apical layers. The adult eyes derived from these discs were only slightly smaller than control eyes, and they occasionally show small scars in the posterior region (Fig 2N and 2O).

Altogether, our data suggest that after inducing cell death in the retinal region, a nonautonomous response is activated, which increases the number of glial cells located in the eye disc epithelium.

## Glial cell proliferation increases in response to apoptotic induction in the retinal cells

The increased number of glial cells observed in damaged eye discs may be due to over-migration of these cells from the optic stalk and/or an excess of glial proliferation. To distinguish between these possibilities, we next examined the proliferation pattern of the glial cells in *GMR-Gal4 tub-Gal80$^{ts}$ UAS-rpr* eye discs after inducing cell death during 72 hours. We observed that upon cell death induction, the proportion of glial cells in S phase was higher than in age-matched control animals, as assayed by 5-Ethynyl-2-deoxyuridine (EdU) incorporation (S2A–S2B" and S2E Fig). In addition, we observed an overall increase in the number of dividing glia upon damage (S2F Fig). Only glial cells located in the basal layer of the eye disc undergo mitosis, as we do not find cells expressing PH3 outside this level.

Next, we analysed the proliferation dynamics of glial cells during recovery time. To this end, we induced cell death in the retina region during a 24-hour period and then determined the number of positive PH3 glial cells at various time points post damage induction. We found that at T0 glial cell proliferation was already elevated compared to control discs. After 24 hours of recovering (T1), glial proliferation was still higher than in undamaged discs, but similar to

that observed at T0. However, at T2 (after 48 hours of recovering), the ratio of glial proliferation was similar to that of control undamaged discs (S3E Fig), suggesting that the signals that promote glial division cease after 48 hours of apoptotic induction.

To evaluate the contribution of cell proliferation on the increased number of glial cells observed in damaged discs, we blocked glial proliferation after inducing apoptosis in the retinal region. To this end, we used the *QF/QUAS* system [21,22] in combination with the Gal4 system. To induce genetic ablation, we expressed *QUAS-rpr* [23] in the retinal cells under the control of *GMR-QF*, whereas glial proliferation was simultaneously blocked by expressing a constitutively activated form of the Retinoblastoma Factor (*UAS-rbf^{CA280}*) under the control of *repo-Gal4* [24]. *GMR-QF>rpr tub-Gal80^{ts} repo>rbf^{CA280}* larvae were raised at 17°C and then shifted to the restrictive temperature to block cell division for 24 hours. After this time, we found that glial cell proliferation was strongly reduced in both damaged and undamaged discs (Fig 3F). Accordingly, we detected that the number of glial cells was also sharply reduced in damaged discs (0.0178 ± 0.0004 glial cells/area $\mu^2$ in control damaged discs, $n$ = 24 versus 0.0049 ± 0.001 glial cells/area $\mu^2$ in *GMRQF>rpr repo>rbf^{CA280}* discs, $n$ = 11, $p$ = <0.0001; Fig 3C–3E). However, in these discs, we observed more glial cells than in control undamaged discs in which proliferation was blocked (0.0049 ± 0.001 glial cells/area $\mu^2$ in *GMRQF>rpr repo-Gal4>rbf^{CA280}* versus 0.001 ± 0.00035 glial cells/area $\mu^2$ in *repo>rbf^{CA280}*, $p$ = 0.02; Fig 3E), suggesting that glial over-migration also contributes to the elevated number of glial cells found after inducing apoptosis. The overexpression of *UAS-rbf^{CA280}* for a prolonged period of time (72 hours) totally abolished glial cell proliferation, yet more glial cells were found when compared to undamaged discs in which proliferation was blocked (Fig 3E and 3F).

Taken together, our data suggest that after cell death induction in the retinal region, signals were generated that ultimately promote nonautonomously glial proliferation and migration, resulting in an increase in the number of these cells in the eye disc.

## Morphological changes in glial cells in response to damage

The eye imaginal disc harbours 2 main glial subtypes, the PN and WG cells. We have examined their behaviour in response to the induction of neuronal cell death. To this end, we induced genetic ablation in the retina using the QF-QUAS system and visualised glial cells with *UAS-GFP* and a membrane-tethered form of GFP (*UAS-mCD8GFP*) under the regulation of *Mz97-Gal4* and *c527-Gal4* driver strains, which drive expression in wrapping and PN glial cells, respectively.

Apoptotic induction in the retinal cells causes a remarkable change in the behaviour and morphology of WG cells. In control discs, WG send long cellular projections that follow the axons through the optic stalk towards the brain. However, in damaged *GMR-QF; UAS-GFP Mz97-Gal4; QUAS-rpr* discs, we did not observe these processes (S4 Fig, compare S4A–S4A' with S4B–S4B'). In damaged discs, the density of WG, as well as the proportion of these cells in the eye discs, was strongly reduced compared to control discs (0.004 ± 0.00024 number of WG/ area $\mu^2$ in control discs, $n$ = 21 versus 0.0026 ± 9.7e-005 in damaged discs, $n$ = 17, $p$ < 0.0001; Fig 4A–4F' and 4Q–4R). Since WG cells never divide, either in control or damaged discs, the reduction in the number of WG upon apoptotic induction is likely due to the stalling of the process of differentiation of WG from PN glia.

In contrast to control discs where WG cells were always located in the basal layer of the discs, in damaged discs we observed these glial cells in the middle/apical layers of the disc epithelium (Fig 4D–4F'). Moreover, whereas in control discs the anterior most row of WG coincided with the anterior leading edge of the glial field, in damaged discs, WG were located between 5 and 4 rows of glia cells behind this boundary (S4 Fig, compare A with B and E).

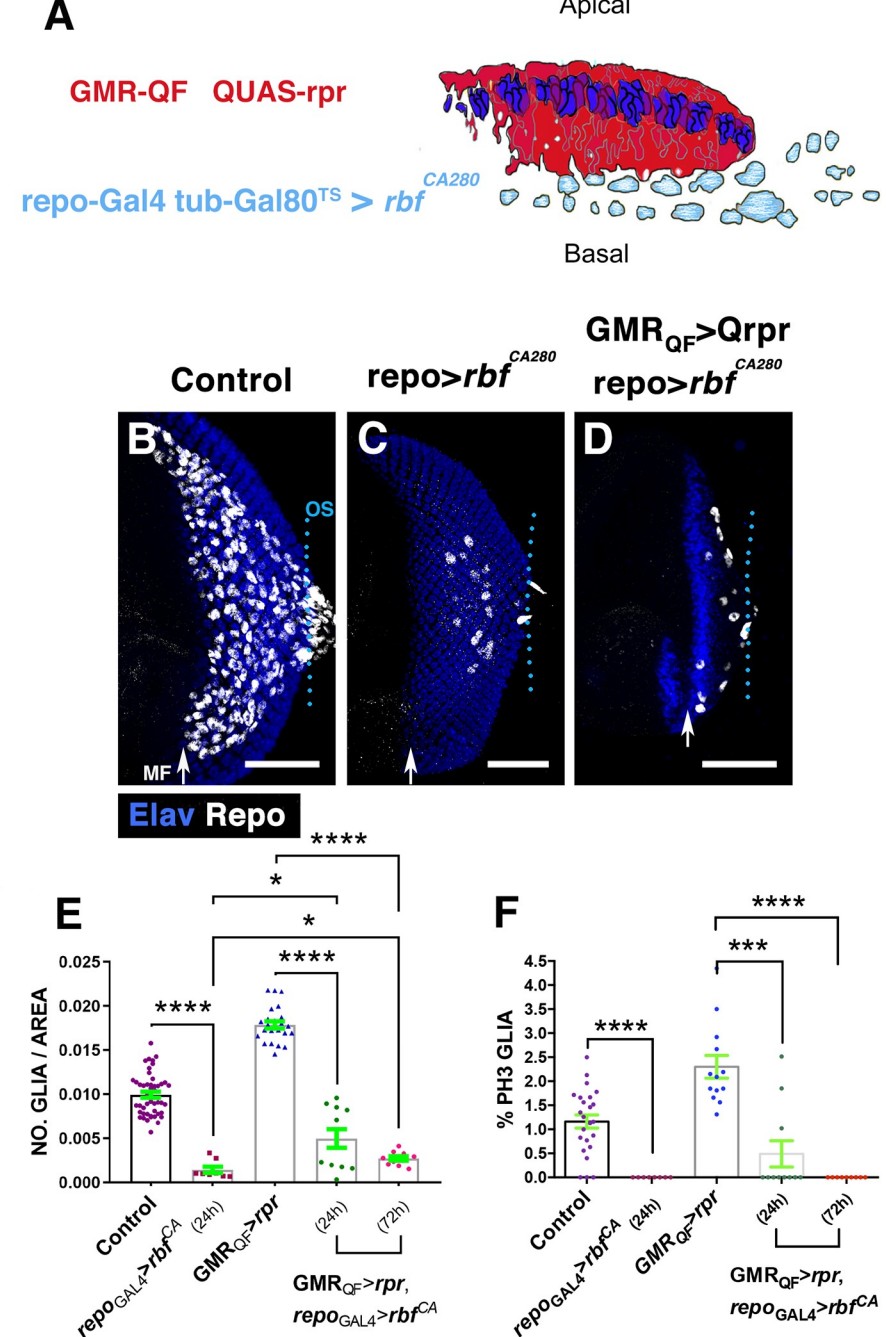

**Fig 3. Overexpression of *rbf*^CA280^ in glial cells reduces the number of glial cells observed after cell death induction.** (A) The schematic illustration represents a transverse section of an eye disc where the region marked in red (expression domain of *GMR-QF*) corresponds to the area of the discs that has been damaged using *QUAS-rpr GMR-QF*. Glial cells are indicated in light blue. *repo-Gal4* drives the expression of *UAS-rbf*^CA280^ under the control of *UAS* specifically in glial cells. (B–D) Third instar eye discs stained with anti-Repo (white) and anti-Elav (blue). Control undamaged *GMR-QF; tub-Gal80*^ts^ *repo-Gal4* eye disc (B), *tub-Gal80*^ts^/+; *repo-Gal4/UAS-rbf*^CA280^ (C) and *GMR-QF*; *tub-Gal80*^ts^/+; *repo-Gal4/UAS-rbf*^CA280^ disc (D). The overexpression of *UAS-rbf*^CA280^ under the control of *repo-Gal4* reduces the number of glial cells in undamaged (C) and damaged eye discs (D). (E) The graph represents the glial density of discs shown in B–D (Control, *repo*~Gal4~-*rbf*~24hrs~, *GMR*~QF~-*rpr*, *GMR*~QF~-*rpr repo*~Gal4~-*rbf*~24hrs~ and *GMR*~QF~-*rpr repo*~Gal4~-*rbf*~72hrs~). (F) The graph shows the percentage of glial cells in mitosis (PH3 positive). Statistical analysis is shown in Table C in S1 Text. Error bars represent SEM. Scale bars, 50 μm. The numerical data used in this figure are included in S1 Data. GMR, glass multiple reporter; *rpr*, *reaper*; SEM, standard error of the mean.

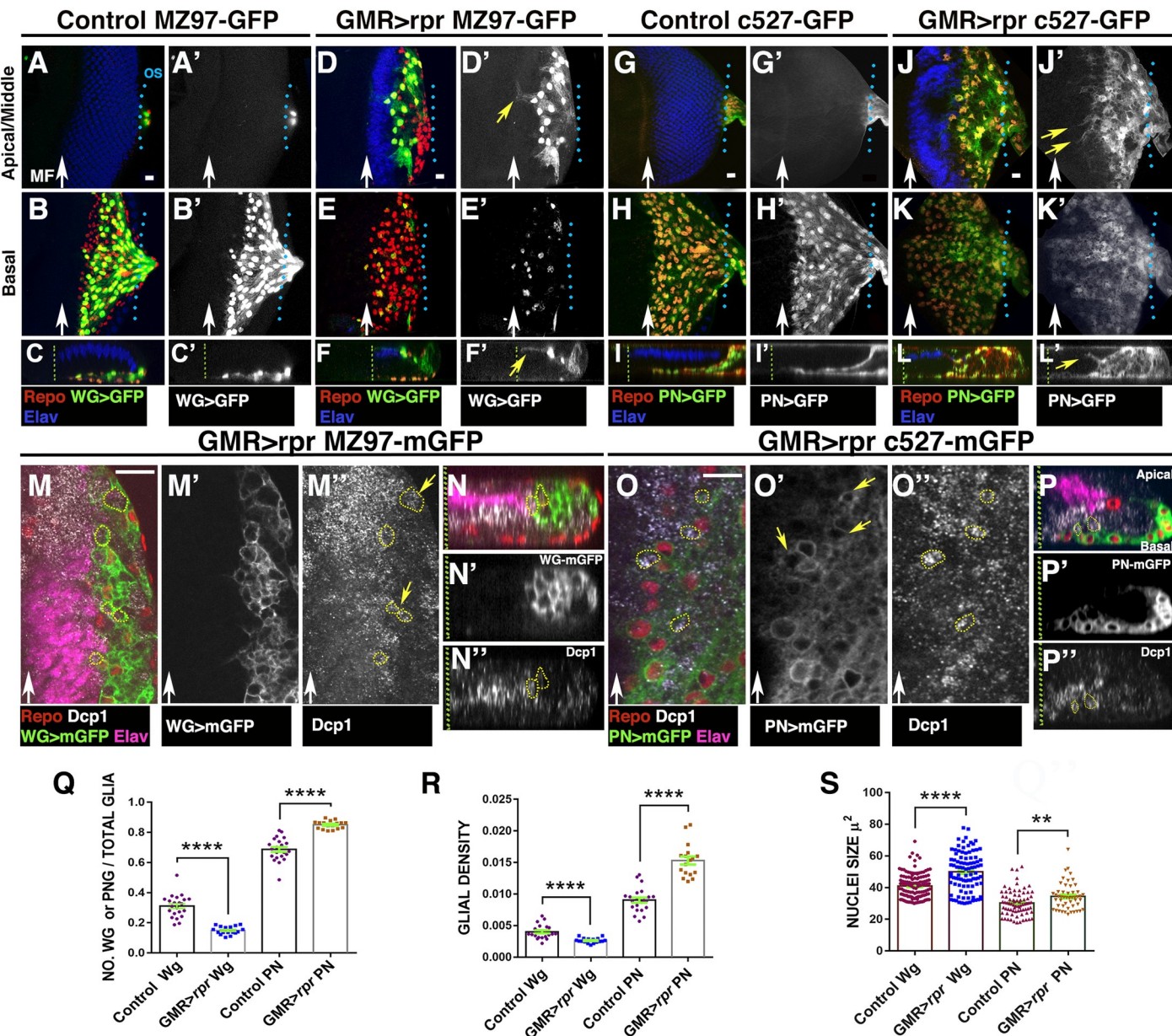

**Fig 4. WG and PN cells change their morphology in response to apoptotic induction in the retinal region.** (A–L) Third instar eye discs stained with anti-Elav (blue) and anti-Repo (red). (A–F') *Mz97-Gal4* driving expression of *UAS-GFP* (green A–C and D–F and grey A'–C' and D'–F') reveals WG cells in control *UAS-GFP Mz97-Gal4; QUAS-rpr* (A–C') and in damaged *GMR-QF; UAS-GFP Mz97-Gal4; QUAS-rpr* (D–F') discs. (G–L') *c527-Gal4* drives expression of *UAS-GFP* (green G–I and J–L and grey in G'–I', J'–L') in PN glial cells in control *UAS-GFP c527-Gal4; QUAS-rpr* (G–I) and damaged *GMR-QF; UAS-GFP c527-Gal4; QUAS-rpr* eye discs (J–L'). (A–A', D–D', G–G', and J–J') Apical/middle layers and (B–B', E–E', H–H,' and K–K') basal layers of the eye disc epithelium. (C–C', F–F', I–I', and L–L') Orthogonal section perpendicular to the furrow through confocal stacks of the eye discs shown in A–K'. (A–F') In control discs, we never find WG cells in the middle or apical layer of the discs (A–C'); however, in damaged discs, most WG cells are located in the apical region (D–F' yellow arrow in F'). These cells extend large processes towards the damaged region (D–D' and F–F'). Some of these projections go over photoreceptors (yellow arrow in F'). (G–L') In damaged eye discs, some PN glial cells were located in the middle layers of the disc epithelium (compared J–L' with control G–I'), and they send cellular projections towards the damage region (yellow arrows in J' and L'). (M–P") Third instar eye discs stained with anti-Elav (magenta), anti-Repo (red), and anti-Dcp-1 (grey). (M–N") *Mz97-Gal4* driving expression of *UAS-mCD8-GFP* (in green and grey in M') reveals WG cells in damaged *GMR-QF; UAS-mCD8-GFP Mz97-Gal4; QUAS-rpr* discs. (O–P") Damaged *GMR-QF; UAS-mCD8-GFP c527-Gal4; QUAS-rpr* eye discs showing the membranes of PN glial cells. Both glial cell types (WG and PN) have vesicles containing cellular debris labelled with anti-Dcp1 (yellow arrows in M" in WG and in O" in PN). (Q–S) Graphs showing ratio of WG and PN glial cells (number WG or PN glial cells/number Total glia) (Q), WG and PN glial density (R), and nuclei size (S). Scale bars, 10 μm. Statistical analysis is shown in Table D in S1 Text. The numerical data used in this figure are included in S1 Data. GMR, glass multiple reporter; PN, perineurial; *rpr*, *reaper*; WG, wrapping glia.

Another interesting feature of WG in damaged discs is the increased complexity of their cytoplasmic projections compared to WG in control discs (Fig 4, compare 4A–4C' with 4D–4F' and S4C–S4C" Fig). Remarkably, we found that some WG cells produce long cellular projections that can extend over the apical surface of photoreceptors (yellow arrows in Fig 4D'–4F' and S4C–S4C" Fig). These results were confirmed by performing a time-lapse image analysis to visualise WG cells in controls and damaged discs. Compared to control discs, we observed that after cell death induction, WG cells extend long membrane projections towards the damaged region (compare control discs S2 Video with S1 Video).

To better characterise the morphology of WG, we visualised their membranes with *MZ97-Gal4 UAS-mCD8-GFP*. We noticed that the end of some of the long cellular projections produced by WG spread out, forming a complex structure (S4C–S4C" Fig). In addition, these glial cells in damaged discs enclose multiple vesicles containing apoptotic bodies, labelled with the apoptotic marker anti-cleaved Dcp-1 (Fig 4M–4N", yellow arrows in 4M" and S4C–S4C" Fig).

The density and proportion of PN glial cells increased upon apoptosis induction (0.009 ± 0.0004 number of PN/area $\mu^2$ in control discs, $n = 21$ versus 0.015 ± 90.00064 in damaged discs, $n = 17$, $p < 0.0001$; Fig 4G–4L", 4Q and 4R). Interestingly, we identified that some PN glial cells located in middle/apical layers produce long cellular projections similar to those described for WG cells (yellow arrows in Fig 4J' and 4L'). We observed that PN glial cells also enclose vesicles containing cellular debris, suggesting that these glial cells have phagocytic activity (Fig 4O–4P"). The average size of the nuclei of PN cells in damaged eye discs is bigger than in undamaged discs Fig 4S.

Our data suggest that WG and PN can engulf and phagocytise cellular debris and apoptotic corpses upon cell death induction in the retinal region. Supporting this, we found that the expression of the microtubule-associated protein light chain 3 (LC3), whose function is required in autophagy and phagocytic processes [25], increases in glial cells of damaged discs (yellow arrowheads in S5B' Fig). We also observed that lysosomal activity strongly increases after cell death induction, as assayed by staining with Lysotracker Red DND-99. In damaged eye discs, glial cells contain multiple organelles labelled with this dye (S5 Fig, compare S5D–S5D' with S5F–S5F"). Accordingly, with a role of glial cells in the process of apoptotic corpses clearance, we observed that the amount of apoptotic debris in the apical region occupied by WG is residual (S4D–S4D" Fig).

Altogether, our data suggest that after cell death induction in the retinal region, some wrapping and PN glial cells expand their membrane surface, generating glial processes. In addition, both glial cells types can engulf and phagocytise apoptotic debris.

## JNK signalling is activated autonomously in damaged regions and nonautonomously in glial cells in response to damage

The JNK pathway plays a fundamental role in regulating many biological processes involved in regeneration, including activation of glial response to damage in the CNS and in the cells anterior to the MF during eye disc development [26–29].

We have examined the activity of this signalling pathway after inducing apoptosis in the retinal region using the JNK reporter (*TRE-GFP*) [30], and a *lac-Z* insertion in the gene *puckered (puc)* [31].

In third instar control eye discs, the *TRE-GFP* reporter is expressed in some photoreceptor cells and unevenly in glial cells (S6A–S6D" Fig). After cell death induction in the retinal region, we observed that the activity of *TRE-GFP* significantly increases in the damaged retina, as well as in some glial cells (S6 Fig, compare S6E–S6H" with control discs S6A–S6D"). We found that

the glial cells that express higher levels of *TRE-GFP* are those located in the middle and apical layers of the eye disc epithelium (blue arrows in S6F'–S6F''' Fig and S6H'–S6H'''). Similar results were obtained with the *puc-lacZ* line upon apoptosis induction (S7 Fig). To further characterise the activity of JNK signalling in glial cells after inducing apoptosis, we analysed the genomic region of *puc* to identify a possible noncoding regulatory region that could be used specifically as a reporter of this signal in glial cells. To this end, we took advantage of the published open chromatin profile of eye discs during tumour development [32]. We selected 2 different regions (*puc-1* and *puc-2*; see S8 Fig) and cloned them into a *lacZ* reporter gene construct. Interestingly, *puc-2* is expressed in glial cells in the eye disc, and a subfragment of this element (named *puc-2B*) reproduced this pattern of expression (S8 Fig, Fig 5). As described for *TRE-GFP*, the reporter *puc2B-lacZ* is expressed unevenly in glial cells of control discs. The proportion of glial cells that express this reporter is similar in both damaged and undamaged control discs (17.39 ± 1.03% in control discs versus 17.26 ± 1.5 in damaged discs; Table E in S1 Text). However, cell death induction leads to an increase in the proportion of glial cells that express *puc2B-lacZ* at high levels (30.13 ± 4.5% of glial cells expressing this reporter in control, $n = 11$ versus 43.91 ± 3.8% of glial cells expressing this reporter in damaged discs, $n = 11$, $p = 0.02$, Table E in S1 Text). The glial cells with higher levels of expression of this reporter are preferentially located in the middle/apical layers of the eye epithelium (Fig 5H–5H" compared to control discs 5D–5D"). Consequently, the proportion of glial cells with elevated levels of *puc2B-lacz* in these layers of the discs was higher than in the basal layer (Fig 5K).

## JNK signal is required for promoting glial migration during glial cell response

The results described so far indicate that in response to damage, JNK signalling was activated in both the damaged region, as well as in glial cells. Therefore, it is plausible that its function may be promoting some aspect of glial response after cell death induction. To complement the expression analysis, we examined glial response after modifying JNK signalling activity in the damaged region or in glial cells.

Firstly, we analysed the autonomous requirement of JNK signalling in the region where cell death was induced by depleting its function in this region. To that end, we co-expressed a dominant negative form of the kinase Basket (*UAS-bsk$^{DN}$*) and *UAS-rpr* under the control of *GMR-Gal4* [33,34]. *UAS-bsk$^{DN}$/UAS-rpr; GMR-Gal4 tub-Gal80$^{ts}$* larvae were transferred during 72 hours at restrictive temperature to induce genetic ablation while overexpressing *UAS-bsk$^{DN}$*. We detected a moderate but reproducible and statistically significant reduction in the density of glial cells present in the eye discs of these larvae (0.019 ± 0.00045 glial cells/μ² in control damaged *GMR>rpr* discs, $n = 39$ versus 0.015 ± 0.0006 glial cells/μ² in damaged *GMR>rpr bsk$^{DN}$* discs, $n = 20$, $p < 0.0001$) (Fig 6D and 6H). The amount of apoptotic cells in these discs was similar to that observed in discs overexpressing *UAS-rpr* exclusively, suggesting that this suppressive effect was not due to attenuation of apoptosis (S9 Fig). Interestingly, we found that there were no statistically significant differences in glial cell proliferation between these discs and control damaged discs (1.75 ± 0.11% mitotic glial cells in control damaged discs versus 1.49 ± 0.37 *GMR>rpr bsk$^{DN}$* discs, $p = 0.89$) (Fig 6I, Table F in S1 Text). The overexpression of *UAS-bsk$^{DN}$* alone (*bsk$^{DN}$; GMR-Gal4 tub-Gal80$^{ts}$*) did not cause any detectable effect on glial cells (Fig 6B and 6H, Table F in S1 Text). These results suggest that the function of the JNK pathway is required in the retina upon apoptotic induction to promote glial response.

We next examined whether the JNK pathway was also required in glial cells. To do this, we combined the *QF/QUAS* system to induce cell death in the retina and the *Gal4/UAS* to block JNK pathway signalling in glial cells under the control of *repo-Gal4* (Fig 6). In *GMR-QF>rpr*

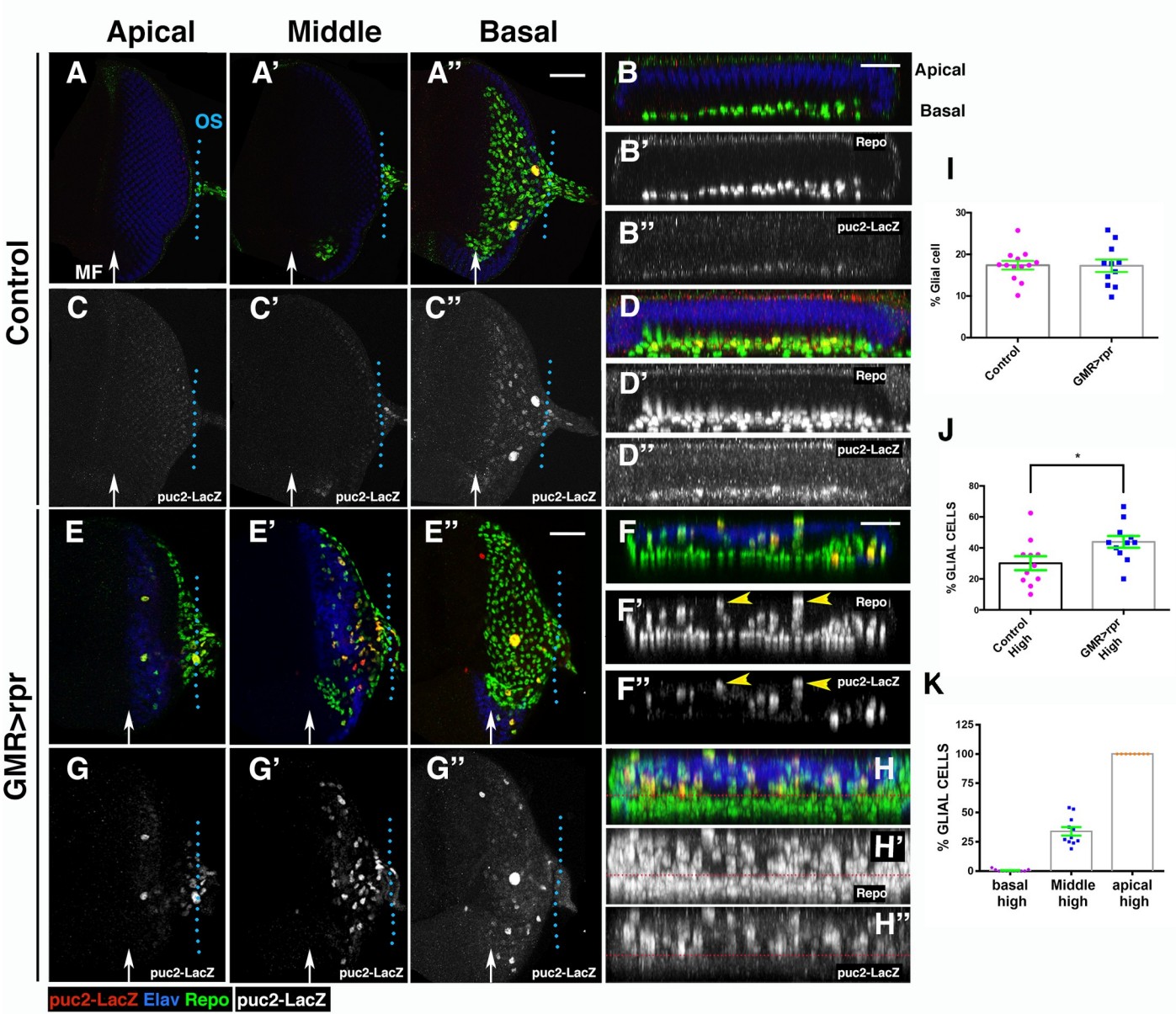

**Fig 5. JNK signalling is activated in response to damage.** (A–H") Third instar eye discs stained with anti-Elav (blue), anti-Repo (green in A–A", B, D, E–E', F, and H and grey in B', D', F', and H') and anti-B-galactosidase (red in A–A", B, D, E–E', F, and H and grey in C–C", B", D", G–G", F", and H") to reveal the activity of the *puc2B-lacZ* reporter, in control *(GMR-Gal4 tub-Gal80^{ts}/+; puc2B-lacZ/+)* (A–D") and damaged *UAS-rpr/+; GMR-Gal4 tub-Gal80^{ts} /+; puc2B-lacZ/+* discs (E–H"). Apical (A, C, E, and G), middle (A', C', E', and G'), and basal (A", C", E", and G") layers of control (A–A" and C–C") and damaged (E–E" and G–G") eye discs. (B–B" and F–F") Y–Z projections show 2 cross sections parallel to the furrow in control (B–B") and damaged (F–F") discs. (D–D" and H–H") Projection of transversal sections generated from 5 different control discs (D–D") and damaged discs (H–H'). Glial cells with higher levels of *puc2B-lacZ* are preferentially located in apical and middle layers of the discs (yellow arrowheads in F' and F"). In the plot shown in H–H", note that most glial cells expressing high levels of *puc2B-lacZ* are located above the red dotted line. (I) Graph shows the percentage of glial cells expressing *puc2-lacZ* (n° cell expressing *puc2B-lacZ*/Total n° of Glia) in control (control) and damaged *UAS-rpr/+; GMR-Gal4 tub-Gal80^{ts}/+; puc2B-lacZ/+* (GMR<rpr) discs. (J) Percentage of glial cells expressing this reporter at high levels (n° cells expressing *puc2B-lacZ* at high levels/total glial cells expressing *puc2B-lacZ*) (see Materials and methods). (K) Graph shows the percentage of glial cells expressing *puc2B-lacZ* at high levels (high) in basal, middle, and apical layers of damaged eye discs. Scale bars, 50 μm. Statistical analysis is shown in Table E in S1 Text. The numerical data used in this figure are included in S1 Data. GMR, glass multiple reporter; JNK, c-Jun N-terminal kinase; *puc*, *puckered*; *rpr*, *reaper*.

*repo-bsk^{DN}* discs, we observed a moderate and statistically significant reduction in the density of glial cells (0.0178 ± 0.0004 glial cells/μ² in control damaged discs, *n* = 24 versus 0.015 ± 0.0004 glial cells/μ² in *GMR-QF>rpr repo-bsk^{DN}* discs, *n* = 21, *p* < 0.0001; Fig 6M and 6N).

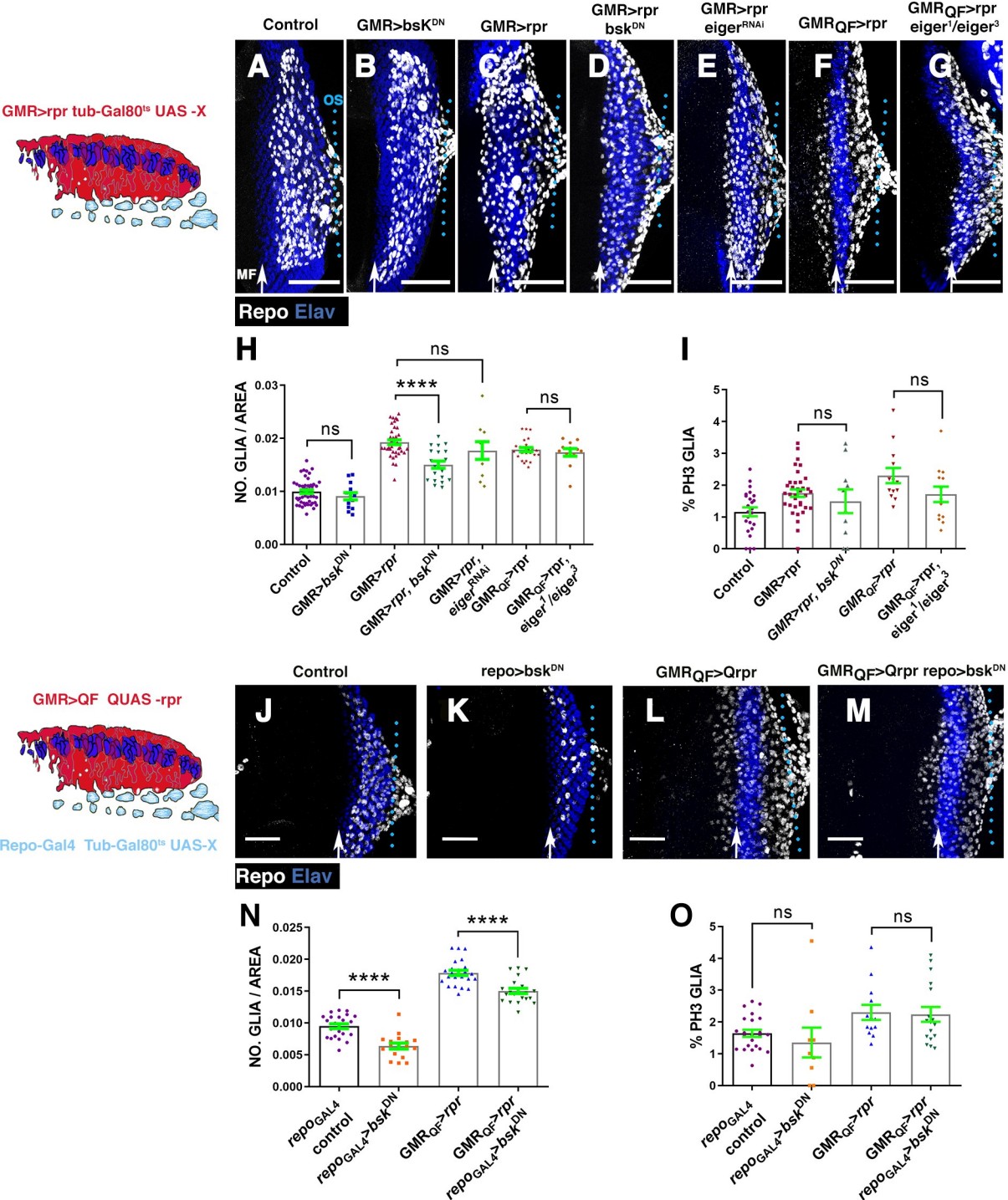

**Fig 6. Reduction of JNK activity prevents the accumulation of glial cells produced upon cell death induction.** (A–G and J–M) Projections of confocal images of third instar eye discs stained with anti-Elav (blue) and anti-Repo (white). The schematic illustration on the left represents a transverse section of an eye disc where the region marked in red (expression domain of *GMR-Gal4*) corresponds to the area of the discs that has been damaged at the same time that JNK was depleted. (A) Control undamaged *GMR-Gal4 tub-Gal80^{ts}/+* disc. (B) *UAS-bsk^{DN}; GMR-Gal4 tub-Gal80^{ts}/+* eye. (C) Damaged *UAS-rpr/+; GMR-Gal4 tub-Gal80^{ts}/+* discs. (D) *UAS-rpr/UAS-bsk^{DN}; GMR-Gal4 tub-Gal80^{ts}/+* damaged discs. (E) *UAS-rpr/; GMR-Gal4 tub-Gal80^{ts}/UAS-eiger^{RNAi}*. (F) *GMR-QF; QUAS-rpr* eye discs damaged. (G) Damaged *GMR-QF; eiger^{1}/eiger^{3}; QUAS-rpr*. All larvae were subjected to the same temperature shifts (see Materials and methods) except larvae of the discs shown in F and G that were raised at 25°C throughout the development. (H–I) Graphs show glial cell density (H) and percentage of mitotic glia cells of discs shown in A–G. (J–M) Effects of the down-regulation of JNK in glial cells of damaging eye discs. In the schematic illustration on the left is

indicated in red the region that has been damaged by overexpressing *QUAS-rpr* under the control of *GMR-QF* and in light blue glial cells. (J) Control undamaged *tub-Gal80^ts^ repo-Gal4* disc. (K) *UAS-bsk^DN^ /+; tub-Gal80^ts^; repo-Gal4/+* eye discs. (L) Damaged *GMR-QF; tub-Gal80^ts^; repo-Gal4 QUAS-rpr* eye discs. (M) *GMR-QF UAS-bsk^DN^; tub-Gal80^ts^; repo-Gal4 QUAS-rpr* damaged discs. (N and O) Graphs show glial cell density (N) and percentage of mitotic glial cells (O) of discs shown in J–M. Scale bars, 50 μm. Statistical analysis is shown in Table F in S1 Text. The numerical data used in this figure are included in S1 Data. GMR, glass multiple reporter; JNK, c-Jun N-terminal kinase; RNAi, RNA interference; *rpr*, *reaper*.

However, the proliferation rate of glial cells in these discs was comparable to that observed in control discs upon cell death induction (2.3±0.23% mitotic glial cells in control damaged discs versus 2.24±0.23 in *GMRQF>rpr repo>bsk^DN^* discs, *p* = 0.57; Fig 6O). The overexpression of *UAS-bsk^DN^* in glial cells reduced the density of glial cells, but did not affect glial cell proliferation (Fig 6N and 6O).

These results suggest that the JNK pathway is required in both the damaged retina as well as in glial cells to increase glial cell number in response to apoptotic induction. However, JNK signalling appears to be dispensable for inducing glial proliferation, suggesting that this pathway is instead necessary to promote glial cell motility. In an attempt to better understand the function of JNK signalling in regulating glial cells behaviour during development, we examined the pattern of proliferation and motility of glial cells in a *hemipterous* (*hep*) mutant background, which encodes a serine/threonine protein kinase involved in the transduction of JNK signalling. In *hep^r75^* mutant eye discs, the total number of glia in the eye disc was significantly reduced, but again without affecting glial proliferation (S10 Fig, Table D in S2 Text). To obtain an overview of glial motility in *hep^r75^* mutant discs, we performed a time-lapse image analysis where glial cells were visualised with *UAS-GFP* driven by *repo-Gal4*. Glial cells migrate by generating short cellular projections spreading towards the leading edge. We observed these projections in both control and *hep^r75^* mutant discs (S11 Fig, S3 and S4 Videos). However, unlike in control discs where glial cells move forward steadily, in *hep^r75^* mutant discs, most glial cells remained in the same position, with only few cells moving slightly forward (S11 Fig, S3 and S4 Videos). Therefore, the average speed and average displacement of glial cells in *hep^r75^* mutant discs were significantly reduced compared with control discs (S11 Fig).

Altogether, our data suggest that JNK signalling is required in both the damaged region, as well as in glial cells, to promote their motility. Our results also imply that after cell death induction in the retina, a signal is produced that nonautonomously activates JNK signalling in glial cells. The tumour necrosis factor (TNF) ligand, termed "Eiger" in Drosophila, is an obvious candidate for being this effector, leading us to examine the effects of the loss of *eiger* on glial response. We first blocked *eiger* activity in the damaged region by co-expressing *UAS-eiger^RNAi^* and *UAS-rpr* under the control of *GMR-Gal4*. The density and proliferation of glial cells in these discs were similar to that found in control damaged discs (Fig 6H). We obtained the same result when analysing these parameters in *eiger^1^/eiger^3^* mutant eye discs after inducing cell death (*QMR-QF; eiger^1^/ eiger^3^; QUAS-rpr*) (Fig 6H). These results suggest that Eiger was not the signal emitted after cell death induction to activate JNK in glial cells. This is supported by our observation that the overexpression of *UAS-eiger* under the control of *GMR-Gal4* only leads to the up-regulation of *TRE-GFP* reporter autonomously, in the GMR domain, but not in glial cells (S12 Fig).

To investigate whether the activation of JNK signalling was sufficient for increasing the number of glial cells in the eye disc, we next examined the consequences of transiently expressing a constitutively activated form of Hemipterous (*hep^CA^*). The ectopic expression of *UAS-hep^CA^* under the control of *GMR-Gal4* causes a moderate but reproducible and statistically significant increase in glial density compared to control discs (S13I Fig). As previously reported by others [35,36], we found that in discs overexpressing *GMR-Gal4 UAS-hep^CA^*, apoptosis

strongly increases. Therefore, the augmented number of glial cells found in these discs might be a consequence of cell death induction. To further examine this idea, we analysed the effects of co-expressing *UAS-hep*$^{CA}$ and *UAS-RHG microRNA* (miRNA). This transgene generates miRNAs that simultaneously inhibit the function of the proapoptotic genes, *rpr*, *hid*, and *grim* [37]. In eye discs co-expressing *UAS-hep*$^{CA}$ and *UAS-RHG* under the control of *GMR-Gal4*, the excess of glial cells found upon overexpression of *UAS-hep*$^{CA}$ was suppressed (S13D and S13I Fig), suggesting that the augmented number of glial cells caused by *UAS-hep*$^{CA}$ was in fact due to the induction of apoptosis.

We then analysed if the activation of *UAS-hep*$^{CA}$ increases glial cell number within eye discs. *tub-Gal80*$^{ts}$; *repo-Gal4/UAS-hep*$^{CA}$ larvae were shifted to 29°C for 24 hours. The number and localisation of glial cells presented in these discs were comparable to those observed in control discs, suggesting that the activation of JNK signalling in glial cells was not sufficient to induce their over-migration into the eye disc (S13 Fig).

## Hh and Dpp signalling are ectopically activated in damaged eye discs

The diffusible factors Dpp and Hh play an important role during the development of eye discs. Hh is known to induce the expression of the BMP2/4 type morphogen Dpp ahead of the MF. This diffusible factor functions redundantly with Hh in MF progression [38–42]. Both Hh and Dpp were also suggested to regulate glial cell development in the eye disc [19,43,44]. Thus, we wondered if they might be involved in the activation of glial response to eye disc damage.

We visualised Dpp expression using an antibody (anti-Dpp) that recognises the Dpp prodomain [45]. After cell death induction, we observed groups of cells behind the MF that express higher levels of Dpp than the surrounding cells (yellow arrowheads in Fig 7D' and 7E'). The levels of expression of some of these cells are similar to those observed in the band of cells anterior to the MF, which can be used as an internal control, since GMR is not expressed in that region (compare plot profiles of control discs Fig 7A" and 7B" with the plot profiles of *GMR>rpr* discs; Fig 7D" and 7E"). Higher magnifications show that some of these cells are photoreceptors (Elav positive, arrowheads in Fig 7F' and 7F"" and green arrow in S14B' Fig). We also find some interommatidial cells with high level of Dpp (yellow arrowhead in S14B' and S14B"' Fig). Similar results were observed when a transcriptional readout for *dpp* expression (*dpp-lacz*) was used (S14C–S14H' Fig).

We further analysed the activity of Dpp signalling pathway upon cell death induction by examining the staining of a phosphorylated form of the transcriptional effector Mad (pMad) [46] and the expression of the Dpp signalling target reporter *dad-lacZ* reporter (*daughters against dpp)* [47]. In control discs, pMad staining was restricted to a band of cells adjacent to the MF, and its expression decreased posteriorly (S14I–S14K Fig). However, in damaged discs, pMad was observed at high levels in most of the cells posterior to the MF, most notably in some glial cells (arrowheads in S14M–S14N' Fig).

In undamaged control eye discs, *dad-lacZ* is expressed from 4 to 5 cell diameters anterior of the MF backwards to the posterior of the disc (Fig 7G–7I'). It is also weakly expressed in all the glial cells of the eye discs. Remarkably, upon cell death induction, the expression of *dad-lacZ* strongly increases in most glial cells of the disc (compare glial cells in control discs Fig 7H–7I', 7M and 7M' with damaged discs 7K–7L', 7N and 7N').

Next, we analysed *hh* expression upon damage induction by examining the activity of an *hh-lacZ* enhancer trap line and the expression of a Hh:GFP tagged line in *UAS-rpr; GMR-Gal4 tub-Gal80*$^{ts}$ eye discs.

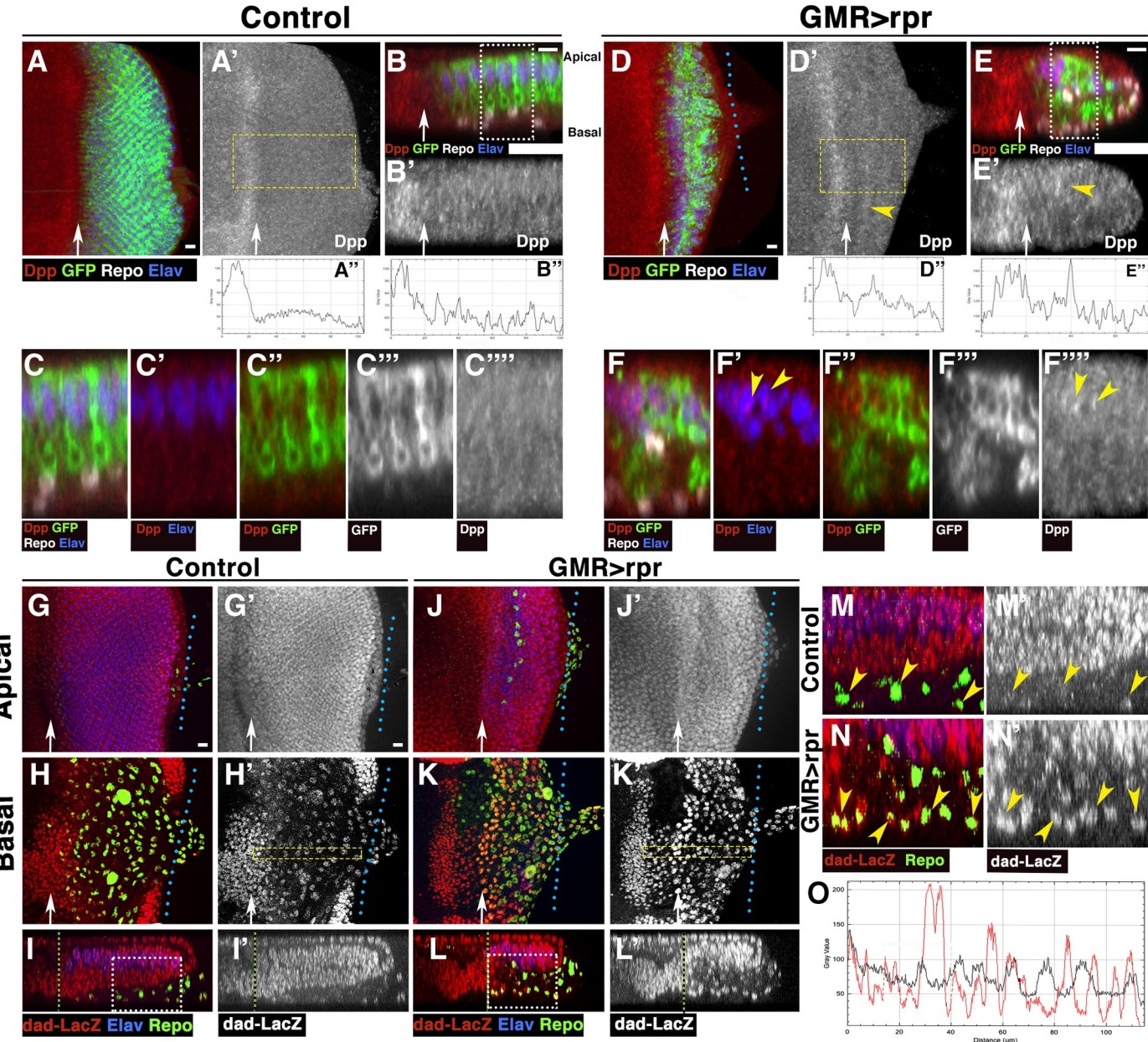

**Fig 7. *Dpp* expression increases after inducing apoptosis in eye discs.** (A–F'''') Third instar eye discs stained with anti-Elav (Blue), anti-Repo (white in A–C and D–F), *GMR-Gal4 UAS-mCD8-GFP* (green in A–C, C'', D–F, and F'' and grey in C''' and F'''), and anti Dpp (red in A–C, C', C'', D–F, F', and F'' and grey in A', B', C'''', D', E', and F''''), in control *tub-Gal80ts GMR-Gal4 UAS-mCD8-GFP* (A–C'''') and damaged *UAS-rpr/+; GMR-Gal4 tub-Gal80ts/UAS-mCD8-GFP* discs (D–F''''). (B–B' and E–E) Cross sections perpendicular to the furrow of the eye discs shown in A (B–B') and D (E–E'). (A'' and D'') Profile plot of the average Dpp intensity across the x-axis of the region highlighted by yellow rectangles shown in panels A' (A'') and D' (D''). (B'' and E'') Profile plot of the average Dpp intensity of the panels shown in B' (B'') and E' (E''). (C–C'''' and F-F'''') Higher magnification images corresponding to the regions highlighted by white rectangles on the panels B (C–C'''') and E (F-F''''). (A–B'') In control eye discs, Dpp is expressed in a band of cells just anterior to the MF, and its expression levels decrease abruptly to a homogeneous low level behind the furrow. (D–E') In damaged discs, some photoreceptors (positive staining for anti-Elav in blue, yellow arrowheads in F') express higher levels of Dpp than the surrounding cells (yellow arrowheads in F' and F''''). In the basal region of the epithelia, where most of the dead cells are located, we only detect low levels of Dpp expression. (G–N') Third instar eye discs stained with anti-Elav (blue), anti-Repo (green), and anti-B-galactosidase (red in G–N and grey in G'–N') to reveal the activity of the *dad-lacZ* reporter in control *(GMR-Gal4 tub-Gal80ts/+; dad-lacZ/+)* (G–I' and M–M') and damaged *UAS-rpr/+; GMR-Gal4 tub-Gal80ts/+; dad-lacZ/+* discs (J–L' and N–N'). (G–G' and J–J') Apical layers of the eye disc epithelium. (H–H' and K–K') Basal layers of the eye disc epithelium. (I–I' and L–L') Y–Z projections show 2 cross sections perpendicular to the furrow in control (I–I'') and damaged (L–L') discs. (M–M' and N–N') Higher magnification images corresponding to the regions highlighted by white rectangles on the panels I (M–M') and L (N–N'). Note that the levels of expression of *dad-lacZ* in glial cells of damaged discs (yellow arrowheads in N and N') are much higher than in control discs (yellow arrowheads in M–M'). (O) Profile plot of the average β-galactosidase intensity across the x-axis of the region highlighted by yellow rectangles shown in panel H' (control discs, black line) and K' (damaged discs, red line). Arrow indicates the approximate position of the MF. The levels of expression of *dad-lacZ* are similar in control and damaged discs in the region anterior to

the MF, but in glial cells posterior to the MF, this reporter is expressed at higher levels in glial cells of damaged discs than in control glial cells. Scale bars, 10 μm. Dpp, Decapentaplegic; GMR, glass multiple reporter; MF, morphogenetic furrow; *rpr*, *reaper*.

In control discs, *hh-lacZ* was expressed at low levels in photoreceptors (S15A–S15D''' Fig). However, in damaged discs, the activity of this reporter was up-regulated in photoreceptor cells (positive for Elav, green arrows in S15H and S15H' Fig).

Hh:GFP is observed in an anterior–posterior gradient and is strongly accumulated in the apical region of the photoreceptors (S16A–S16F'' Fig). Upon apoptosis induction, we find high levels of Hh:GFP throughout the eye discs, not only in the region close to the MF (S16G–S16I' Fig). Hh:GFP accumulates forming large aggregates in the middle and basal layers of the epithelium, in contrast to control discs, where Hh:GFP always is detected at high levels in the apical region (S16 Fig, compare S16G–S16J''' with control S16A–S16F''). Most of these aggregates are adjacent to photoreceptors and are not positively stained for DAPI (S16 Fig yellow arrowheads in S16I', and S16J' and S16J''), indicating that they do not correspond to cells expressing high levels of Hh, but are likely produced by the adjacent photoreceptors. Occasionally, we find pyknotic nuclei, labelled with DAPI, that have high levels of Hh:GFP (S16 Fig yellow arrows in S16J–S16J'), suggesting that Hh:GFP is also detected at high levels in some apoptotic cells. Accordingly, we find some cells with high levels of Hh:GFP that were also positive for Dcp-1 (blue arrowheads in S16K–S16L'' Fig). Interestingly, we observed some glial cells engulfing Hh:GFP aggregates (yellow arrowheads in S16M–S16N'' Fig).

We next examined whether increased levels of *hh* expression in retinal cells might activate Hh signalling in subretinal glia by analysing the expression of *patched* (*ptc*), a known target of this signalling pathway. Consistent with our previous results, we observed that in *GMR>rpr* eye discs the levels of Ptc were abnormally high in cells posterior to the MF, as well as in subretinal glial cells (green arrow in S15P–S15P' Fig).

These data suggest that after inducing cell death in the retinal region, some apically located photoreceptor neurons and some interommatidial cells up-regulate Dpp and Hh in response to apoptotic signals. These factors, in turn, nonautonomously would activate these 2 signalling pathways in glial cells.

## Dpp and hh signalling are necessary to promote glial response

Our observations indicate that in response to cell death induction, both Dpp and Hh signalling were activated in the retina region, as well as in glial cells, leading us to consider if these signals might be involved in stimulating glial response to damage. To assess this, we first examined whether the depletion of *dpp* and *hh* would alter the density and/or proliferation of glial cells observed in damaged discs. To this end, we performed RNA interference (RNAi)-mediated knockdowns of Hh and Dpp levels by expressing RNAi lines under the control of *GMR-Gal4* while simultaneously inducing apoptosis using *UAS-rpr*. None of the *UAS-dpp*$^{RNAi}$ lines employed in our assay significantly modified the effects on glia numbers observed after apoptotic induction (Fig 8C and 8G, S17 Fig). We obtained similar results when Hh levels were knock down upon induction of cell death using the same experimental setup (Fig 8D and 8G).

Next, we tested whether the simultaneous depletion of Dpp and Hh levels affects glial response by co-expressing *UAS-dpp*$^{RNAi}$, *UAS-hh*$^{RNAi}$ and *UAS-rpr* under the control of *GMR-Gal4*. We found that Dpp and Hh simultaneous knockdown impairs the accumulation of glial cells upon cell death induction (0.019 ± 0.00045 glial cells/μ$^2$ in control *GMR>rpr* discs, $n = 39$ versus 0.015 ± 0.00039 glial cells/μ$^2$ in *GMR>rpr UAS-dpp*$^{RNAi33}$ *UAS-hh*$^{RNAi}$ discs, $n = 18$, $p < 0.0001$; Fig 8A, 8B and 8G). Accordingly, we found that compared to control damaged discs, glial cell proliferation was reduced (1.8 ± 0.1% mitotic glial cells in control

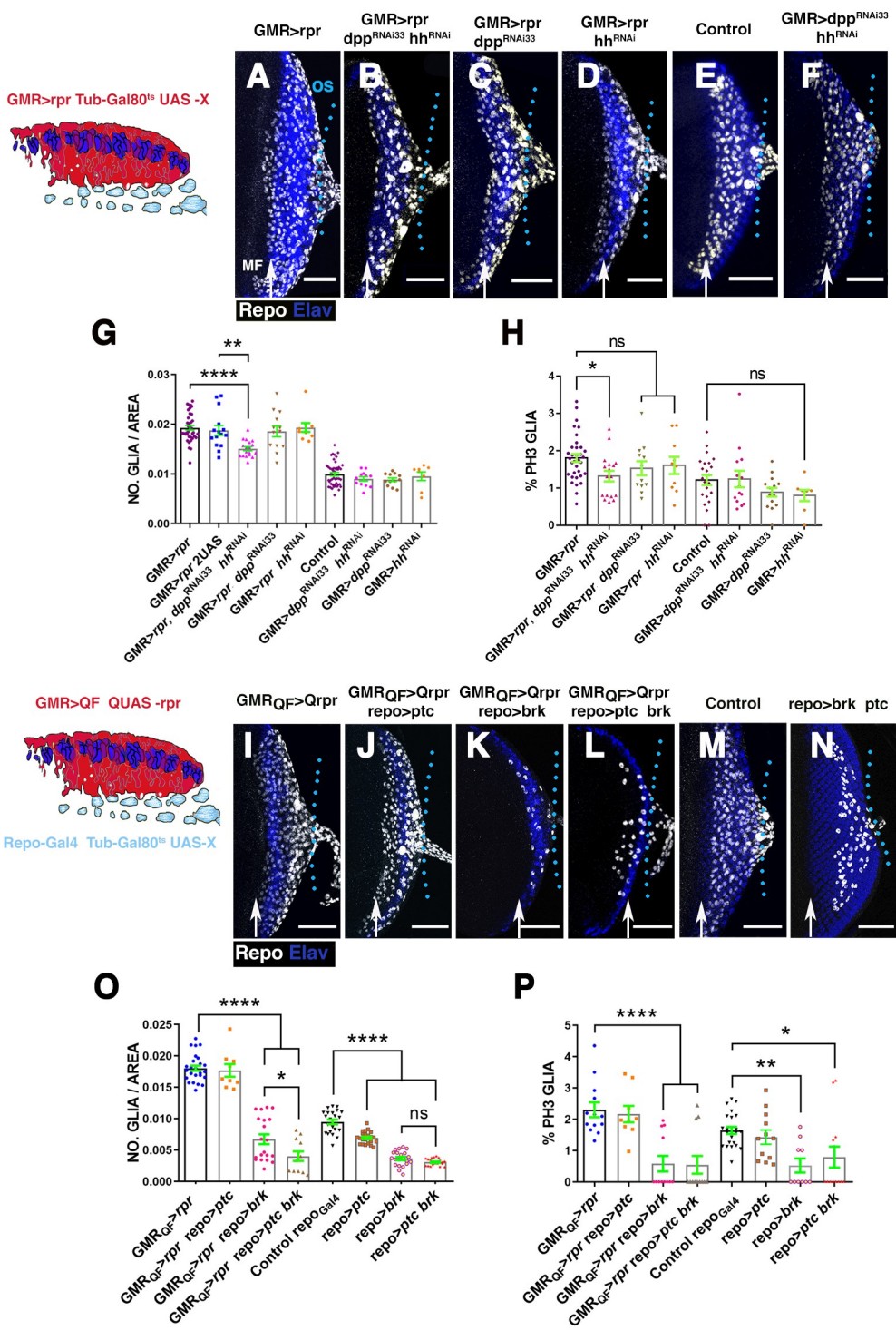

**Fig 8. The down-regulation of Dpp and Hh signalling reduce glial response.** (A–F and I–N) Projections of confocal images of third instar eye discs stained with anti-Elav (blue) and anti-Repo (white). (A–F) Effects caused by the down-regulation of Dpp and Hh signalling in the damaged region. The schematic illustration on the left represents a transverse section of an eye disc where the region marked in red (expression domain of *GMR-Gal4*) corresponds to the area of the discs that has been damaged at the same time that Dpp and/or Hh signalling were depleted. (A) *UAS-rpr/+; GMR-Gal4 tub-Gal80^ts^/+* damaged eye disc. (B) Damaged *UAS-rpr/+; GMR-Gal4 tub-Gal80^ts^/ UAS-hh^RNAi^; UAS-*

$dpp^{RNA33i}$. (C) Damaged *UAS-rpr/+; GMR-Gal4 tub-Gal80$^{ts}$; UAS-dpp$^{RNAi33}$* (BDSC33618). (D) *UAS-rpr/+; GMR-Gal4 tub-Gal80$^{ts}$/ UAS-hh$^{RNAi}$* damaged discs. (E) Control undamaged *GMR-Gal4 tub-Gal80$^{ts}$/+* disc. (F) *GMR-Gal4 tub-Gal80$^{ts}$/ UAS-hh$^{RNAi}$; UAS-dpp$^{RNAi33}$* eye discs. (G and H) Graphs show glial cell density (G) and percentage of mitotic glia cells of discs shown in A–F. (I–N) Effects of the down-regulation of Dpp and Hh signalling in glial cells in damaged discs. In the schematic illustration on the left is indicated in red the region that has been damaged (*GMR-QF; QUAS-rpr*) and in light blue glial cells. (I) *GMR-QF; tub-Gal80$^{ts}$; repo-Gal4 QUAS-rpr* eye disc. (J) *GMR-QF; tub-Gal80$^{ts}$/UAS-ptc; repo-Gal4 QUAS-rpr*. (K) *GMR-QF; tub-Gal80$^{ts}$; repo-Gal4 QUAS-rpr/UAS-brk*. (L) *GMR-QF; tub-Gal80$^{ts}$/UAS-ptc; repo-Gal4 QUAS-rpr/UAS-brk*. (M) Control undamaged *tub-Gal80$^{ts}$; repo-Gal4* disc. (N) *tub-Gal80$^{ts}$/UAS-ptc; repo-Gal4 /UAS-brk*. (O and P) Graphs show glial cell density (O) and percentage of mitotic glial cells (P) of discs shown in I–N. Scale bars, 50 μm. Statistical analysis is shown in Table G in S1 Text. The numerical data used in this figure are included in S1 Data. GMR, glass multiple reporter; Dpp, Decapentaplegic; Hh, Hedgehog; RNAi, RNA interference; *rpr*, *reaper*.

*GMR>rpr* discs versus 1.23 ± 0% mitotic glial cells in *GMR>rpr UAS-dpp$^{RNAi33}$ UAS-hh$^{RNAi}$* discs, *p* = 0.01; Fig 8H). The down-regulation of *dpp* and *hh* function under the control of *GMR-Gal4* using the same experimental conditions neither affect the density or the division of glial cells in control conditions (Fig 8G and 8H, Table G in S1 Text). To confirm that this effect was not due to a titration of Gal4 activity caused by introducing an additional transgene, we used damaged eye discs that have 2 *UAS* reporters as a control (Fig 8G). In this experimental setup, the effects on glial density were similar to those found in control damaged discs with only 1 *UAS* construct (Fig 8G).

We next examined the consequence of inducing apoptosis in the retinal region and simultaneously blocking Dpp and/or Hh pathways in glial cells. Once again, we combined the QF/QUAS and Gal4/UAS system. To block Dpp signalling in glial cells, we overexpressed *brinker* (*brk*) or *dad*, under the control of *repo-Gal4*. *brinker* encodes for a sequence-specific transcriptional repressor that negatively regulates Dpp-dependent genes, whereas *dad* encodes the inhibitory SMAD in the BMP/Dpp pathway and subsequently down-regulates Dpp signalling activity [48].

The overexpression of *UAS-brk* or *UAS-dad* in the glial cells of damaged discs was sufficient to prevent the accumulation of glia induced upon damage (0.018 ± 0.0004 glial cells/μ$^2$ in control damaged discs, *n* = 27 versus 0.0067 ± 0.007, glial cells/μ$^2$ in *GMR-QF>rpr repo>brk*, *n* = 20 *p* < 0.0001, and 0.01 ± 0.0037 glial cells/μ$^2$ in *GMR-QF>rpr repo>dad*, *n* = 13 *p* < 0.0001; Fig 8O, S17 Fig). Also, glial proliferation decreased in damaged discs expressing *UAS-brk or UAS-dad* (2.3 ± 0.23% mitotic glial cells in *GMR-QF>rpr repo-Gal4* versus 0.58 ± 0.25% in *GMR-QF>rpr repo>brk* discs, *p* < 0.0001 and 1.07 ± 0.29 in *GMR-QF>rpr repo>dad*, *n* = 11 *p* = 0.009; Fig 8P). The overexpression of these genes under the control of *repo-Gal4* without inducing apoptosis also caused a significant reduction in the number of glial cells, as well as in their rate of proliferation (Fig 8O and 8P, S17 Fig, Table G in S1 Text, Table F in S2 Text). These results imply that Dpp signalling pathway is necessary for increasing the number of glial cells in response to damage.

We also analysed the effects of blocking Hh signalling in glial cells upon apoptotic induction, by overexpressing *UAS-ci$^{RNAi}$* or *UAS-ptc*, under the control of *repo-Gal4*. We observed that after inducing cell death, the down-regulation of Hh signalling did not cause any significant change in the number of glial cells compared to control damaged eye discs (Fig 8J, 8O and 8P, S17 Fig, Table G in S1 Text, Table F in S2 Text).

In line with our previous observations, we found that upon apoptotic induction, the simultaneous depletion of Hh and Dpp pathways in glial cells by co-expressing *UAS-dad* and *UAS-ci$^{RNAi}$* or *UAS-brk* and *UAS-ptc* under the control of *repo-Gal4*, significantly reduced the number of glial cells, compared with discs in which only Dpp signalling was depleted (0.0067 ± 0.007 glial cells/μ$^2$ in *GMR-QF>rpr repo>UAS-brk*, *n* = 20 versus 0.004 ± 0.0007 glial cells/μ$^2$ in *GMR-QF>rpr repo>-ptc brk* discs, *n* = 12, *p* = 0.047; Fig 8N, 8O and 8P,

S17 Fig, Table G in S1 Text, Table F in S2 Text). Interestingly, the proliferative ratio of the glial cells in these discs was similar to that found when only Dpp signalling was reduced, suggesting that Dpp plays a more important role in the control of glial proliferation than Hh signalling.

We next examined the consequence of overexpressing *UAS-dpp* and *UAS-hh* under the control of *GMR-Gal4*. Excess glial cells were observed after overexpressing *dpp*, but not when *hh* was ectopically expressed ($0.0099 \pm 0.00034$ glial cells/$\mu^2$ in control discs, $n = 47$ versus $0.0122 \pm 0.00045$ glial cells/$\mu^2$ in *GMR>dpp* discs, $n = 15$, $p = 0.0039$; S18E Fig, Table G in S2 Text). The ectopic expression of both genes produce a significant increase in glial density with respect to that observed in *GMR>dpp* discs ($0.0122 \pm 0.00045$ glial cells/$\mu^2$ in *GMR>dpp* discs, $n = 15$ versus $0.0145 \pm 0.00079$ glial cells/$\mu^2$ in *GMR>dpp hh* discs $n = 14$, $p = 0.046$; S18 Fig, Table G in S2 Text). Glial cells proliferation also increased after the co-overexpression of both genes, but not when they are individually overexpressed (S18 Fig, Table G in S2 Text). To complement this analysis, we examined the effects caused by the ectopic activation of these signalling pathways in glial cells. To this end, we ectopically expressed under the control of *repo-Gal4* a constitutively activated version of the receptor Thickvein ($tkv^{QD}$), *Interference hedgehog* (*Ihog*) and a mutated form of *ci* ($ci^{m1\text{-}3^{*}\,103}$) that increases the activity of this transcription factor [49–52]. *ihog* encodes for a transmembrane protein that is essential for Hh pathway activation [53]. We confirmed that the overexpression of *UAS-ihog* was sufficient to up-regulate Hh signalling by observing high Ptc levels in these discs (S19 Fig). The density of glial cells in eye discs overexpressing *UAS-ihog* or *UAS-ci*$^{m1\text{-}3^{*}\,103}$ under the control of *repo-Gal4* was similar to that of control discs (Fig 9G). Whereas the overexpression of *UAS-tkv*$^{QD}$ under the control of repo-Gal4 (*repo-Gal4 UAS-tkv*$^{QD}$) caused a mild increase in glial density ($0.0094 \pm 0.00037$ glial cells/$\mu^2$ in control discs, $n = 23$ versus $0.012 \pm 0.0004$ glial cells/$\mu^2$ in *repo>tkv*$^{QD}$ discs $n = 18$, $p = 0.03$; Fig 9), although glial cell proliferation was not statistically significantly augmented ($1.64 \pm 0.11\%$ mitotic glial cells in control discs, $n = 22$ versus $1.52\pm0.15\%$ in *repo>tkv*$^{QD}$ discs, $n = 16$, $p = 0.9$; Fig 9H). However, the co-overexpression of *UAS-tkv*$^{QD}$ and *UAS-ihog* under *repo-Gal4* strongly increased both glial cell density as well as proliferation ($0.0094 \pm 0.00037$ glial cells/$\mu^2$ and $1.64 \pm 0.11\%$ mitotic glial cells in control discs versus $0.019 \pm 0.001$ glial cells/$\mu^2$ and $2.54 \pm 0.2\%$ mitotic glial cells in *repo>ihog tkv*$^{QD}$; Table H in S1 Text). These effects are not reproduced after co-expressing *UAS-tkv*$^{QD}$ and *UAS-ci*$^{m1\text{-}3^{*}\,103}$ (Fig 9). Interestingly, the coactivation of both signalling pathways, either by co-expressing *UAS-tkv*$^{QD}$ and *UAS-ci*$^{m1\text{-}3^{*}\,103}$ or *UAS-tkv*$^{QD}$ and *UAS-ihog*, promotes the over-migration of glial cells. Thus, whereas in most discs overexpressing either *UAS-tkv*$^{QD}$, *UAS-ihog* or *UAS-ci*$^{m1\text{-}3^{*}\,103}$, the anterior border of glial migration lies 2 to 4 rows of ommatidia posterior to the MF, a high percentage of discs overexpressing simultaneously both *UAS-tkv*$^{QD}$ and *UAS-ihog* or *UAS-ci*$^{m1\text{-}3^{*}\,103}$ shown glial cells overcoming that border (Fig 9D, 9E and 9I).

## Glial motility stimulation by Dpp and Hh signalling depends on JNK function

The results described so far indicate that JNK signalling facilitates glial motility during normal development, as well as in response to apoptotic induction. Similarly, our observations suggest that Dpp and Hh signalling stimulate both the proliferation and the motility of the subretinal glia. Next, we investigated whether JNK mediates some of these functions promoted by Dpp and Hh signalling. Firstly, we examined the activity of *puc2B-lacz* in discs overexpressing *UAS-dpp* and/or *UAS-hh* in the retinal region under the control of *GMR-Gal4*. The ectopic expression of *UAS-hh* did not significantly modify the activity of this reporter (S20B' Fig), while the overexpression of *UAS-dpp* increased the percentage of glial cells that express the reporter ($17.39 \pm 1\%$ glial cells express the reporter in control discs, $n = 13$ versus $25.9 \pm 1.9$ in

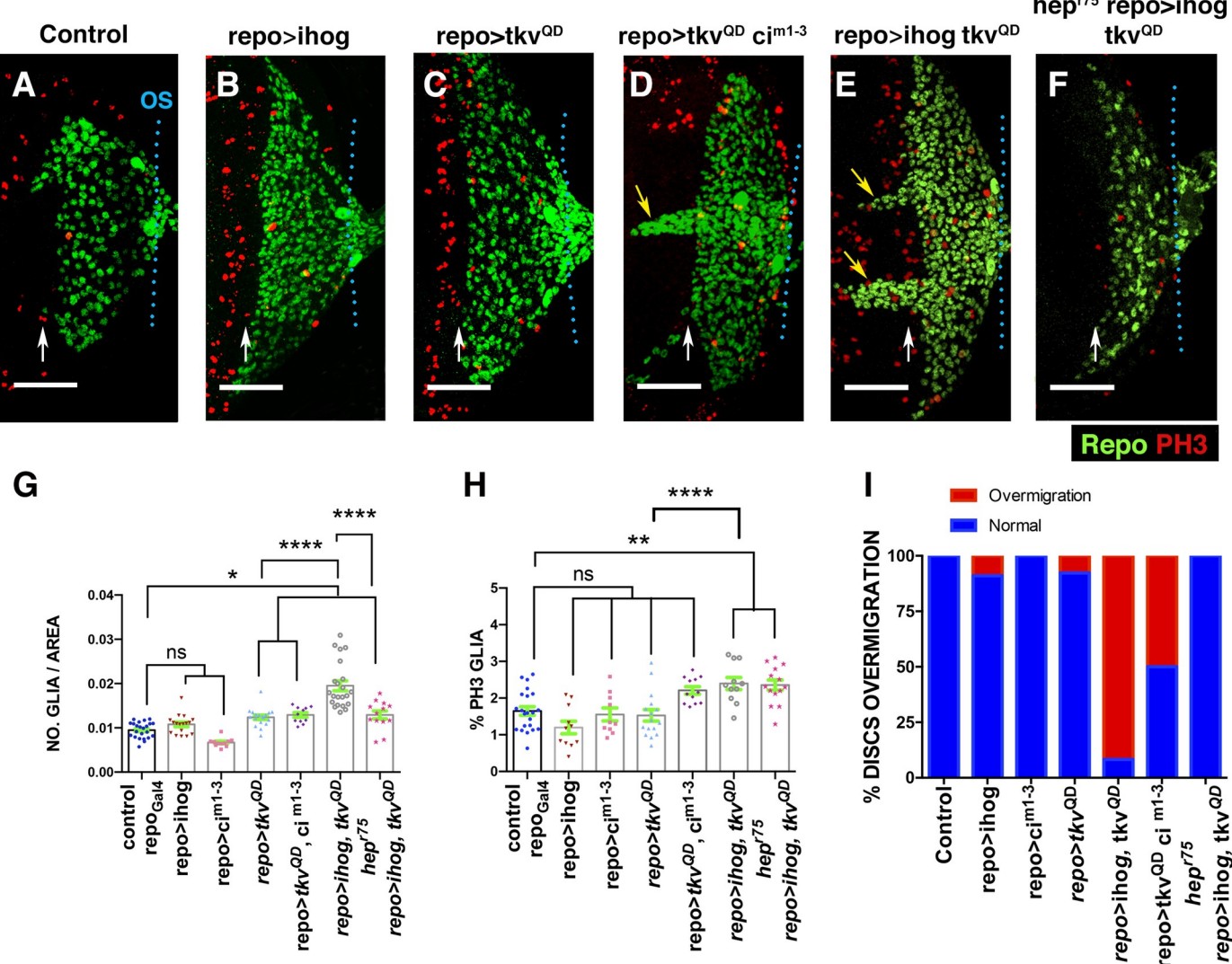

**Fig 9. Overexpression of *ihog* and *tkv*^QD in glial cells induces over-migration and proliferation.** (A–F) Third instar eye discs stained with anti-PH3 (red) and anti-Repo (green). Basal layers of the eye disc epithelium. (A) Control *tub-Gal80^ts repo-Gal4* eye disc. (B)*tub-Gal80^ts/UAS-ihog; repo-Gal4/+*. (C) *tub-Gal80^ts/+; repo-Gal4/UAS-tkv^QD*. (D) *tub-Gal80^ts/UAS-tkv^QD/UAS-ci^m1-3; repo-Gal4/UAStkv^QD*. (E) *tubGal80^ts/UAS-ihog;repo-Gal4/UAS-tkv^QD*. (F) *hepr75; tub-Gal80^ts/UAS-ihog; repo-Gal4/UAS-tkv^QD*. The overexpression of *tkv^QD* under the control of *repo-Gal4* increases the number of glial cells (C). The co-overexpression of *UAS-tkv^QD* and *UAS-ihog* under *repo-Gal4* increases the density and proliferation of subretinal glial cells and promotes the over-migration of these cells (E). The over-migration phenotype and the increase of glial density caused by the overexpression of *UAS-tkv^QD* and *UAS-ihog* with *repo-Gal4* was suppressed in the *hep^r75* mutant discs. Nonetheless, the glial cell division was not significantly altered (F and H). (G) The graph represents the glial density of discs shown in A-F. (H) Histogram showing the percentage of glial cells in mitosis (Ph3 positive) of discs indicated in A–F. (I) The graph represents the percentage of discs with glial over-migration shown in A–F. Scale bars, 50 μm. Statistical analysis is shown in Table H in S1 Text. The numerical data used in this figure are included in S1 Data.

*GMR>dpp* discs, *n* = 9, *p* = 0.0036; S20C and S20E Fig), as well as the proportion of glial that express the reporter at high levels (5.4 ± 0.991% glial cells express the reporter at high levels in control discs, *n* = 11 versus 11.02 ±1.67 in *GMR>dpp* discs, *n* = 9, *p* = 0.047; S20E Fig). The co-expression of both factors strengthens the latter effect (S20D–S20D" Fig), although the percentage of glial cells expressing *puc2B-lacz* is comparable to that observed when *UAS-dpp* was expressed alone (S20E Fig, Table H in S2 Text). Thus, these results suggest that Dpp signalling is sufficient to activate JNK signal on glial cells, although Hh signalling strengthens this effect.

To define whether JNK signalling mediates some of the effects induced by these 2 signalling pathways, we examined glial proliferation and density in $hep^{r75}$ mutant discs after overexpressing $UAS\text{-}tkv^{QD}$ and $UAS\text{-}ihog$ under the control of $repo\text{-}Gal4$. In this mutant background, there is a drastic reduction in the density of glial cells compared to $UAS\text{-}tkv^{QD}$ $UAS\text{-}ihog$ $repo\text{-}Gal4$ discs (0.019 ± 0.001 glial cells/μ² in $repo{>}tkv^{QD}$ $ihog$ discs, $n = 22$ versus 0.012 ± 0.0009 in $hep^{R75}$ $repo{>}tkv^{QD}$ $ihog$ discs, $n = 13$, $p < 0.0001$; Fig 9, Table H in S1 Text). However, glial cell proliferation was not significantly altered (2.39 ± 0.16% mitotic glial cells in $repo{>}tkv^{QD}$ $ihog$ discs, $n = 11$ versus 2.35 ± 0.13% in $hep^{R75}$ $repo{>}tkv^{QD}$ $ihog$ discs, $n = 14$, $p = 0.9$; Fig 9, Table H in S1 Text). We also found that the over-migration phenotype caused by the overexpression of $UAS\text{-}tkv^{QD}$ $UAS\text{-}ihog$ was totally suppressed in the $hep^{r75}$ mutant background (91.67% in $repo{>}tkv^{QD}$ $ihog$ discs, $n = 18$ versus 0% in $hep^{R75}$ $repo{>}tkv^{QD}$ $ihog$ discs, $n = 18$; compare Fig 9D and 9E with 9F). Therefore, these results are consistent with a model in which the function of JNK is necessary to facilitate the over-migration of glial cells induced when Dpp and Hh signalling are overexpressed.

## Glial activity in response to damage is mediated by JNK signalling

Next, we investigated whether the JNK signalling pathway, which was highly activated in some WG and PN cells in response to damage, influences the behaviour of these cells. We first examined WG morphology and behaviour upon damage induction in different mutant conditions of the JNK signalling pathway.

To specifically knock down JNK function in WG cells, we overexpressed $UAS\text{-}puc$ or $UAS\text{-}bsk^{DN}$ under the regulation of $Mz97\text{-}Gal4$. In undamaged $Mz97\text{-}Gal4$ $UAS\text{-}GFP$ $UAS\text{-}puc$ discs the density, proportion (WG/Total glial cells), morphology, and localisation of WG were not altered when compared with control discs ($Mz97\text{-}Gal4$ $UAS\text{-}GFP$). However, in discs ectopically expressing $bsk^{DN}$ under the control of $Mz97\text{-}Gal4$ or in $hep^{r75}$ mutant discs, we observed a weak, but statistically significant reduction of the density of WG (Fig 10K), although the proportion of these cells was not altered (Fig 10L). The down-regulation of JNK signalling in WG cells, either via the overexpressing $UAS\text{-}puc$ or $UAS\text{-}bsk^{DN}$, in damaged discs altered neither the density of these cells nor their location in the apical/middle layers of the discs (Fig 10G–10I' and 10K). However, we did detect a reduction in the proportion of WG (WG/total glial cells) in these discs compared to injured control discs (Fig 10L). Moreover, although we observed cellular debris inside these WG cells (S21 Fig), they hardly ever developed cytoplasmic projections as those presented in activated WG in control discs (compare Fig 10E with 10H). This is most clearly seen in a time-lapse image analysis (compare S2 Video with S5 Video). Another feature of activated WG cells was the enlargement of their nuclei. Interestingly, this effect was suppressed when JNK was reduced ($49.9{\pm}1.28$ μ² in $GMR{>}rpr$ $Mz97\text{-}GFP$ discs versus $39.5{\pm}1.56$ in $GMR{>}rpr$ $Mz97{>}bsk^{DN}$ discs, $p < 0.0001$; Fig 10M).

We also examined whether the reduction of the function of Hh and Dpp signalling affects the ability of WG to engulf cellular debris. We observed that WG overexpressing either $UAS\text{-}brk$ or $UAS\text{-}ptc$ under the regulation of $Mz97\text{-}Gal4$ in damaged discs contain cellular debris, as assayed by staining for anti-cleaved Dcp-1 (S21 Fig).

## Cell death induction in the leg disc epithelium promotes the accumulation of glial cells

To determine whether the signalling network regulating glial response in the eye discs functions in other regions of the PNS, we examined glial cells response upon apoptotic induction in the leg discs. The leg imaginal disc is an epithelial structure that comprises a variety of sense organs arranged in a precise and reproducible pattern. These sensory elements contain glial

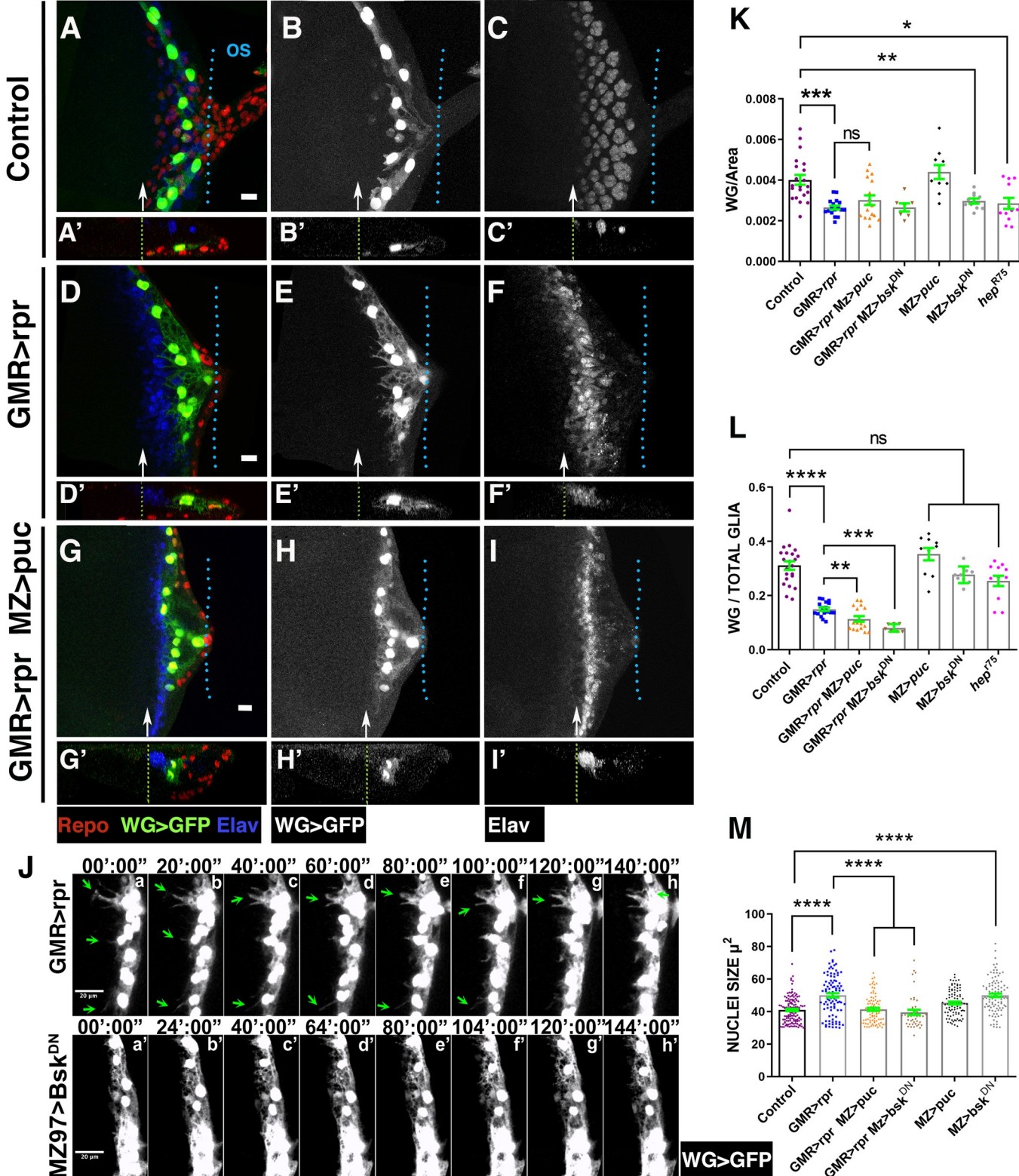

**Fig 10. JNK depletion affects the morphology and behaviour of WG in damaged eye discs.** (A–I') Third instar eye discs stained with anti-Elav (blue A, A', D, D', G, and G' and grey C, C', F, F', I, and I') and anti-Repo (red). (A–I') *Mz97-Gal4* driving expression of *UAS-GFP* (green A, A', D, D', G, and G' and grey B, B', E, E' H, and H') reveals WG cells in control *UAS-GFP Mz97-Gal4; QUAS-rpr* (A–C), damaged eye discs (*GMR-QF; UAS-GFP Mz97-Gal4; QUAS-rpr (*D–F'), and damaged eye discs with depleted JNK signalling function *GMR-QF; UAS-GFP Mz97-Gal4; repo-Gal4 QUAS-rpr/UAS-puc* (G–I'). (A'–C', D'–F', and G'–I')

Cross sections perpendicular to the furrow of eye discs shown in A–I. The apical localisation of WG cells observed in damaged discs is not prevented by the reduction of JNK signalling (compared E' to H'). However, when the function of JNK signalling is blocked, WG cells do not produce the large processes found in WG in damaged disc (compared E to H). (J) Detailed frames from in vivo time-lapse analysis (S5 Video) of eye discs. When JNK is blocked, WG cells do not form cellular projections, as it occurs in damaged discs. (K–M) Graphs showing WG and PN glial density (K), ratio of WG and PN glial cells (number of wrapping or PN glial cells/total number of glial cells) (L), and nuclei size (M). Scale bars, 10 μm. Statistical analysis is shown in Table I in S1 Text. The numerical data used in this figure are included in S1 Data. GMR, glass multiple reporter; JNK, c-Jun N-terminal kinase; PN, perineurial; *puc*, *puckered*; *rpr*, *reaper*; WG, wrapping glia.

cells that are specified in the CNS/PNS transition region during larval development and have to migrate through a nerve into the forming leg (Fig 11A and 11B) [16].

To analyse glial cells response upon apoptotic inductions in the leg disc, we transiently overexpress *UAS-rpr* under the control of *Distal-less (Dll)* or *hh-Gal4*. *Dll-Gal4* is expressed in the central part of the leg, in a region corresponding to the distal elements of the appendages, and is not active in glial cells (Fig 11, yellow arrows in 11F). The overexpression of *UAS-rpr* during 48 hours leads to increased levels of apoptosis throughout the *Dll* domain (Fig 11G'). The number of glial cells in the leg epithelium after inducing apoptosis strongly increases compared with control discs (102.2 ± 4.07 number of glial cells in control discs, $n = 60$ versus 153.1 ± 11.67 in *Dll>rpr*, $n = 15$, $p < 0.0001$; Fig 11C and 11G–11G''', Table J in S1 Text). Interestingly, in contrast to control discs in which glial cells are located close to the nerve branches innervating the leg discs [16] (Fig 11I–11J''), in damaged leg discs, some glial cells leave the nerve and spread over the leg epithelium (Fig 11, compare 11I–11J'' with 11K–11L''). In contrast to eye discs, we do not find that glial cells proliferation increases upon apoptosis induction (Fig 11D). Similar results were observed when *rpr* was expressed under the *hh-Gal4* line (Fig 11C and 11D).

We next examined the activity of JNK signalling after inducing apoptosis. In third instar control leg discs, the *TRE-GFP* reporter is expressed in a group of cells in the distal segment of the appendage and at low levels in few glial cells (S22 Fig). Upon cell death induction using *Dll-Gal4*, the activity of *TRE-GFP* significantly increases throughout the discs epithelium, as well as in glial cells (yellow arrowheads in S22C–S22D' Fig). To complement the expression analysis, we examined glial response to cell death induction in a *hep^{r75}* mutant background. The total number of glial cells was significantly smaller in *hep^{r75} hh>rpr* damaged leg discs than in control *hh>rpr* damaged leg discs (96.91 + 8.8 glial cells in *hep^{r75} hh>rpr leg discs* versus 163±18 in *hh>rpr*, $p = 0.0001$; S22E–S22I Fig, Table I in S2 Text).

We also examined whether the expression of *dpp* and *hh* was altered upon apoptosis induction in the leg disc. The overexpression of *UAS-rpr* under the control of *Dll-Gal4* induces the ectopic expression *hh* throughout the central region of the leg disc, including the anterior compartment (S23 Fig). However, Hh signalling was not activated in glial cells, as assayed by staining for anti-Ptc (S23G–S23L' Fig). The expression of *dpp* was also altered upon apoptotic induction, since we observed that in these discs *dpp-lacZ* is expressed throughout the domain of expression of *Dll-Gal4* (S24 Fig compare S24B–S24B' with control discs S24A–S24A'). Accordingly, the expression of *dad-lacZ* was also altered in the leg discs epithelium (S24 Fig). Interestingly, *dad-lacZ* was detected also in glial cells that enter in the leg epithelium (S24 Fig blue arrows in S24H and S24H').

To assess whether Dpp and or Hh signalling might be involved in stimulating glial response in the leg disc, we first examined whether the depletion of *dpp* and *hh* affects glial response in leg dics. To do this, we overexpressed *UAS-hh^{RNAi}* and/or a *UAS-dpp^{RNAi33}* during 48 hours under the control of *Dll-Gal4* while simultaneously inducing cell death using *UAS-rpr*. The overexpression of *UAS-dpp^{RNAi33}* impairs the accumulation of glial cells in leg discs after cell death induction (146.2 ± 5.94 glial cells in *Dll>rpr*, $n = 25$ versus 105 ± 7.54 glial cells in

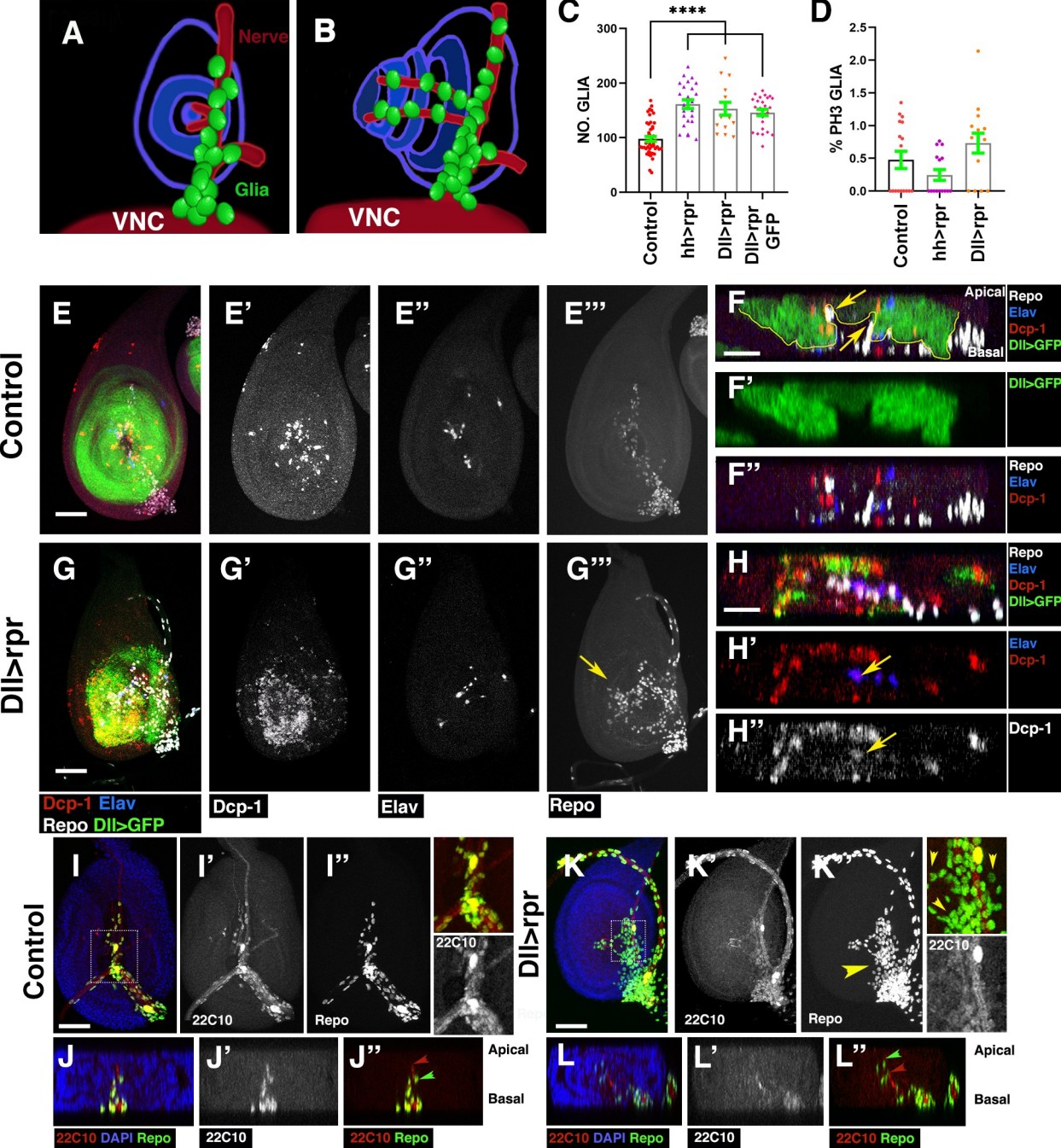

**Fig 11. The induction of cell death in the leg disc causes glial cells accumulation in the leg disc epithelium.** (A and B) Schematic illustrations of leg discs, frontal view (A) and lateral view (B). Leg discs are concentrically organised, and they are connected to the VNC through the leg nerve. In third instar larvae, the glial cells in the leg discs (in green) are accumulated along the leg nerve (in red) and along 2 nerves into the telescoping leg. (C) The graph indicates the total number of glial cells in control and leg discs after cell death induction using *Dll-Gal4* and *hh-Gal4* lines. (H) Histogram showing the percentage of glial cells in mitosis (PH3 positive) in leg discs analysed in C. (E–E"" and G–G"") Projections of confocal image stacks of control *Dll-Gal4 tub-Gal80^{ts}/UAS-mCD8-GFP* (E–E"'), and *UAS-rpr*; *Dll-Gal4 tub-Gal80^{ts} / UAS-mCD8-GFP* (G–G"") leg discs after inducing cell death. (F–F" and H–H") X–Z projections show a cross section of the leg discs epithelium shown in E (F–F") and G (H–H"). The discs were stained for anti-Repo (white in E, E"', F, F", G, G"', and H) anti-Dcp1 (red in E, F, F", G, H, and H' and grey in E', G', and H"), anti-Elav (blue in E, F, F", G, H, and H' and grey in E" and G"), and *Dll>CD8GFP* (green in E, F, F', G, and H). *Dll-Gal4* is not expressed in glial cells, since these cells do not express GFP under the control *Dll-Gal4* (yellow arrows in F). The overexpression of *UAS-rpr* under the control of *Dll-Gal4 tub-Gal80^{ts}* induces cell death throughout the leg discs epithelium, including some of the neurons forming part of the sense organs contained in this structure (yellow arrows in H' and H"). (I–I" and K–K") Projections of confocal image stacks of control *Dll-Gal4 tub-Gal80^{ts}* (I–I") and *UAS-rpr*; *Dll-Gal4 tub-Gal80^{ts}* (K–K") leg discs after inducing cell death. (J–J" and L–L") Cross section of the leg discs shown in I (J–J") and K (L–L"). The leg

discs were stained for anti-Repo (green in I, J, J", K, L, and L" and grey in I" and K"), DAPI (blue in I, J, K, and L), and anti-22C10 (red in I, J, J", K, L, and L" and grey in I', J', K, and L'). Higher magnification images corresponding to the regions highlighted by white rectangles on the panels K and I are shown on the right panels. Note that whereas in control leg discs glial cells are located close to the nerve branches of the leg discs, upon apoptosis induction, glial cells detached from the nerve and spread throughout the leg disc (yellow arrowhead in K and yellow arrowheads in the high magnification'), compared undamaged discs (I–J") with damaged discs (K and L"). Compare also the position of glial cells (green arrowhead in J" and L") with respect to the nerve (red arrowhead in J" and L") in control discs and damaged discs. In control discs, glial cells are located along the nerves into the telescoping leg (J"); however; upon cell death induction, some glial are detached from the nerve and appear distally to this (L"). Scale bars, 50 μm. Statistical analysis is shown in Table J in S1 Text. The numerical data used in this figure are included in S1 Data. *rpr*, *reaper*; VNC, ventral nerve cord.

*Dll>rpr dpp$^{RNAi33}$*, $n = 17$, $p = 0.0001$; S25F and S25J Fig). However, glial cells proliferation was not altered in these discs (S25J and S25K Fig). We obtained similar results when cell death was induced in a *dpp$^{d12}$/dpp$^{d14}$* mutant background (S25H–S25J, Table J in S2 Text). In contrast to the results obtained in the eye discs, the number of glial cells in damaged leg discs in which Dpp and Hh levels were simultaneously depleted were similar to those found in discs in which only the Dpp pathway was down-regulated (S25G and S25J Fig, Table J in S2 Text).

Accordingly, with a function of Dpp signalling promoting glial response in leg discs, we observed that a high number of glial cells accumulate in the leg discs after overexpressing *UAS-dpp* under the control of *Dll-Gal4* (190.6 ± 11.37 glial cells in *Dll>dpp* discs, $n = 22$ versus 102.2 ± 4.07 control discs, $n = 60$, $p < 0.0001$; S26B and S26H Fig, Table K in S2 Text). The overexpression of *hh* did not cause any detectable change in the number of glial cells in the leg discs (S26C and S26H Fig, Table K in S2 Text). We did not find that glial cells proliferation increases after overexpressing either *UAS-dpp* or *UAS-hh*. The ectopic expression of both genes simultaneously produced an increase in the number of glial cells similar to that seen in *Dll>dpp* leg discs (S26D and S26H Fig). In concordance with these results, we find that the number of glial cells in leg discs expressing *UAS-tkv$^{QD}$* under the control of *repo-Gal4* strongly increases compared with control discs (102.2 ± 4.07 glial cells in control discs, $n = 60$ versus 199.5±16.36 in *repo>tkv$^{QD}$*, $n = 13$, $p < 0.0001$, Table K in S2 Text, S26F Fig). Surprisingly, in this genetic background, the proportion of mitotic glial cells was slightly augmented compared to control discs (S26K Fig). The overexpression of *UAS-ihog* did not significantly alter the number of glial cells compared to control discs (S26E Fig.

All together, our results indicate that in leg discs, as it occurs in eye discs, the function of Dpp and JNK signalling pathways is involved in the glial response that is triggered in response to apoptotic induction.

## Discussion

The eye disc contains several glial cells that overall have equivalent functions to some of the mammalian glial cells found in the PNS [2,7,54]. The unique developmental features of these discs have made them an ideal model in which to establish the molecular underpinnings of glial development and function [7,8,9,11]. However, the use of the eye disc as a model system to study glial response to neuronal damage and the mechanisms that might be regulating them has remained largely unexplored. Glial regenerative response (GRR) is found across many species and may reflect a common underlying genetic mechanism [1,55]. Considering that *Drosophila* glial cells have served as an experimental model to gain insights into mammalian glial biology, eye discs might provide an excellent model system to discover evolutionarily conserved signalling networks regulating glia regenerative response.

We used the eye disc to explore the mechanisms involved in promoting glial cell response to the induction of neuronal apoptosis during development. Our results show that in eye discs, glial cells response is controlled by the activity of Dpp and Hh signalling pathways. The

function of these pathways is necessary for stimulating the proliferation and migration of glial cells to the eye discs after apoptosis induction. JNK pathway would be mediating this latter process.

Different reports have shown that during normal development Hh and Dpp signalling pathways play an important role in the development of eye discs, as well as regulating glial cell development in this structure [43,44]. Therefore, it is possible that the mechanism that we described in this model system might be exclusively activated in the eye discs as a compensatory developmental response. To examine this possibility, we have analysed whether this mechanism was involved in glial response in other regions of the PNS. Remarkably, we found that a similar mechanism controls glial response upon apoptosis induction in the leg discs. As it occurs in the eye discs, leg discs glial cells are specified in the CNS and migrate into the forming leg. Cell death induction in leg disc epithelium promotes glial cells accumulation in these structures. In contrast to the eye discs, we did not observe an excess of mitotic glial cells, suggesting that glial cells proliferation is not mediating this effect. However, in our analysis, we only have examined whether glial cell proliferation increases in the leg discs, but not in the leg nerve or in the CNS/PNS transition zone, where leg glial cells are specified; hence, we cannot rule out that glial proliferation might be involved in increasing glial cells in response to apoptotic induction in the leg disc.

Our data indicate that the function of Dpp and JNK, but not Hh signalling, is required for this glial response. The distinct developmental programme of the leg and eye discs imply that the effects of Dpp and JNK on glial cells are not due to compensatory developmental response of eye cells, and, therefore, suggest that these signalling pathways might be mediating glial response to cell death induction in different regions of the PNS.

The eye disc contains different glial cell types including a specialised PN glial cell type, the WG. This glia type is an axon-associated cell that enwraps axons resembling non-myelinating Schwann cells forming Remak fibers in the mammalian PNS [7]. In these animals, nerve injury triggers the conversion of non-myelin (Remak), as well as myelin Schwann glial to a cell type specialised in promoting repair. During this process, these cells undergo large-scale changes in gene expression and morphology. In mammals, peripheral nerves owe their regenerative potential to the ability of Schwann glial to convert to cells devoted to repair after injury [17,56]. These properties make Schwann cells attractive candidates for regenerative therapies of the injured spinal cord [57,58].

In this work, we have shown that when the neuronal region of the eye disc epithelium was damaged, it induced a glial response that consisted of an increase in glial migration and division. In addition, we observed that WG cells, and some PN glia, undergo morphological changes and acquire phagocytic activity. This behaviour resembles the regenerative response of non-myelinating Schwann cells [17,56,59]. Therefore, the eye disc is a good model to reveal regulatory mechanisms involved in glial response in the PNS, which might be of relevance to mammalian organisms.

There have been described 6 morphologically distinct glial cell classes in the eye discs. In this work, we focused our analysis on the 2 main glial cell types contained in the eye disc. However, in our functional analysis, we have used anti-Repo to characterise the effects on glial numbers of the different mutant conditions examined. Since Repo is expressed in all glial cells contained in the discs, it is possible that other subtype glial cells of the eye discs could contribute to the results we have obtained.

Our data show that upon apoptotic induction, the nucleus size of glial cells increases. Enlarge nucleus can be due to a polyploisation process. In mammals, polyploid cells are present in many organs where they play important roles regulating tissue homeostasis in response to damage [60]. Somatic polyploidy can arise through different mechanisms, one of them is via

endomitosis. During this process, cells do enter mitosis, but do not undergo cell division, and can result in polyploid cells with a single large nucleus. Changes in the cell cycle of glial cells caused by the signals produced by the damaged region might induce this process increasing the size of nuclei of glial cells.

## Activation of JNK signalling in glia cells is necessary but not sufficient for inducing glial response

In *Drosophila*, the function of the JNK signalling pathway is required for regenerative processes in both the CNS and PNS. In adult and larva brains, the function of Eiger/TNF is involved in triggering glial proliferation upon injury [26]. In the PNS, the JNK signalling pathway is activated in response to axon injury in damaged neurons, where it regulates injury-induced transcriptional changes [61]. The activation of this signalling pathway in the damaged neuron can promote axonal regrowth [62,63]. In addition, it has been shown that axon self-destruction leads to the activation of Draper/Ced-1/MEGF-10. This protein is an engulfment receptor that promotes clearance of cellular debris in different organisms, including *Drosophila* [64]. In glial cells, Draper binds to TRAF4 and Shark. This promotes cytoskeletal rearrangements (a process essential for phagocytic activity), and, at the same time, induces the activation of JNK signalling, leading to the induction of dAP-1-mediated gene transcription, including Draper. The activation of this factor in glial cells facilitates glial engulfment of axonal debris [28].

The involvement of JNK in promoting other aspects of GRR, such as glial proliferation and motility in the PNS remained largely unexplored. The data presented in this work suggest that the function of JNK signalling is required in both the damaged region, as well as in subretinal glial cells to promote glial motility but not proliferation. Our results also indicate that the activation of JNK in glial cells in response to damage was not mediated by the ligand Eiger. An alternative mechanism to explain how JNK signalling might be activated in the damaged eye is via Draper [64].

Damage in peripheral nerves of mammals provokes the activation of the transcription factor c-Jun [56]. This early event in the regenerative response plays a fundamental role in regulating the major aspects of injury response. It determines the formation of regeneration tracks and myelin clearance and is required for efficient Schwann cell dedifferentiation. This later process activates a repair programme and transforms these cells into cells specialised to support regeneration [56]. The repair Schwann cell is a transient cell state required for repairing injured tissue [17]. Similarly, our results suggest that in the eye disc, damage induces the activation of JNK signalling in WG, and, subsequently, these cells undergo various morphological and behavioural changes. We show that the reduction of JNK signalling in transformed WG partially suppresses some of the effects induced by damage, for instance, the formation of the long membranes projection observed in transformed WG was inhibited. However, our data indicate that the ectopic activation of JNK signalling in WG in control undamaged discs was not sufficient for inducing the WG transformation, suggesting that other signal pathways must be involved in regulating this process.

## Dpp and Hh signalling promote JNK activation for facilitating glial motility

Our work reveals that the eye disc is a powerful model system in which to define the mechanisms that control glial cell migration, proliferation, and activity transformation, which occur in response to the induction of cell death in neuronal tissue. The results presented in this work lead us to propose a mechanism describing how the signalling pathways of Dpp and Hh might

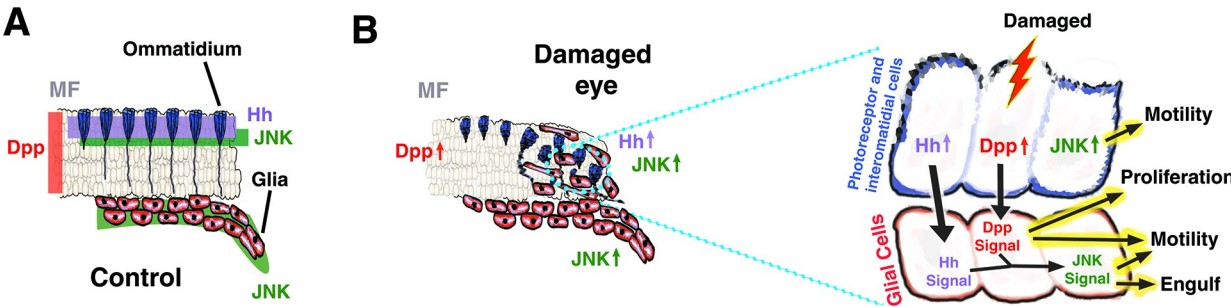

**Fig 12. Model representing the signalling network involved in the activation of glial response after cell death induction in eye discs.** (A) In the eye discs, Hh is expressed in the interommatidial and photoreceptors cells, behind the MF (purple). The expression of Dpp is located ahead of the MF (red). JNK is expressed in the glial cells and the retina (green). (B) Damage promotes a GRR consisting of increased glial migration and division. Moreover, it is stimulated the engulfment activity of glial cells for removing cellular debris and apoptotic corpses. When cell death is induced in the retinal tissue, the Hh and Dpp pathways induce glial cell proliferation and migration and activate the JNK pathway that facilitates glial migration and engulfment. The basal displacement of photoreceptors is likely due to the restructuring of the epithelium caused by apoptotic induction. Dpp, Decapentaplegic; GRR, glial regenerative response; Hh, Hedgehog; JNK, c-Jun N-terminal kinase; MF, morphogenetic furrow.

promote this glial response (Fig 12). We have found that upon apoptotic induction in the retina region, some cells ectopically express Dpp and Hh. Interestingly, we have found that the expression of these genes is not only activated in apoptotic cells, but also in other non-apoptotic cells. These results suggest that apoptotic cells might be generating signals that would induce the expression of *dpp* and *hh*, in the surrounding cells. This, in turn, would induce the nonautonomously activation of both signalling pathways in glial cells to promote their proliferation and induce the activity of JNK pathway to facilitate their motility (Fig 12). Our data indicate that the down-regulation of *dpp* in the injured region in the eye was not sufficient for modifying glial response. Only the simultaneous silencing of *dpp* and *hh* in the damage region altered glial behaviour. This might be due to the weakness of the loss-of-function effects caused by RNAis. Alternatively, both signalling pathways might have a redundant function in promoting glial response, as previously shown during neuronal differentiation in eye discs [40]. In contrast to eye discs, in leg discs, the function of Hh signalling is not mediating this response, and only Dpp signalling is required. This result supports the idea that Dpp is the main signal promoting glial response. Therefore, whereas in other tissues, Dpp would be sufficient for promoting glial response, in the eye discs, because the particular functional relationship between Hh and Dpp signalling in this structure, this function might be at least partially mediated by Hh signalling. It would be interesting to examine whether in other regions of the PNS *hh* and *dpp* can also have a redundant function promoting glial response. The relative mild effects observed after depleting Dpp and Hh signalling in the eye discs suggest that other signal pathways must be involved in regulating this process. The signalling pathways of EGFR and FGF are good candidates for mediating this response. Both signalling pathways affect glial behaviour and at least the function of the EGFR signalling has been associated with stress response in different systems [65].

## Materials and methods

### *Drosophila* stocks and genetics

The following stocks were used:

*UAS* lines: *UAS-GFP (II and III), UAS-mCD8-GFP (III), UAS-dpp (III), UAS-reaper (X), UAS-rpr (II), UAS-20XmCherry* (B52267)(described in Flybase, Bloomington Drosophila Stock Centre) *UAS-dpp^{RNAi33}* (Bloomington Drosophila Stock Centre: 33618), *UAS-dpp^{RNAi2}*

(Kyoto DGGR, 9885R-3), *UAS-dpp$^{RNAiInt}$ (UAS-dppshmiR)* [66], *UAS-hep$^{CA}$ (II) (III)* [67], *UAS-tkv$^{QD}$* [68], UAS-*brk* [69], *UAS-puc2A* [31], *UAS-bsk$^{DN}$* [67], *UAS-hh-GFP* (II) [70], *UAS-ihog* (VDRC stock: 29897), *UAS-hh$^{RNAi}$* (VDRC stock:17874), *UAS-ptc* [71], *UAS-dad* [48] *UAS-ci$^{RNAi}$* (NIG-FLY, 2125R-1), *UAS-ci (m1-3* 103)* (BSC 32571), *UAS-rbf$^{280}$* [72], and *UAS-microRNARHG (II)* [37].

Mutants: *eiger$^1$/Cyo*, *eiger$^3$/Cyo*, *hep$^{r75}$/FM7 dpp$^{d12}$/Cyo*, and *dpp$^{d14}$/Cyo*. All of these stocks have been previously described in FlyBase.

The reporter lines are the following: *puc-lacZ* line [31], *dpp-lacZ*, BAC-encoded Hh>GFP and *hh-lacZ*, (described in Flybase), TRE-DsGFP [30], and *dad-lacZ* [47].

We used the following Gal4 lines: *GMR-Gal4 tub-Gal80$^{ts}$ /CyO (pGMR-Gal4)* [73], *tub-Gal80$^{ts}$; repo-Gal 4/TM6B*, *tub-Gal80$^{ts}$ hh-Gal4/TM6b*, *tub-Gal80$^{ts}$ Dll-Gal4*, *c527-Gal4*, and *Mz97-Gal4* [43].

We used the following QF/QUAS lines: *GMR-QF; tub-Gal80$^{ts}$/Cyo; repo-Gal4/TM6B*, *QUAS-rpr repo-Gal4/TM6b* and *QUAS-rpr /TM6b* [21–23].

Stocks and crosses were maintained on yeast food at 25°C or 17°C before and after the transitory inhibition of *Gal80$^{ts}$* at 29°C.

## Genetic analysis

To analyse the effects on glial cells behaviour after inducing apoptosis in the retinal region, *UAS-rpr*; *GMR-Gal4 tub-Gal80$^{ts}$/CyO*, larvae were raised at 25°C until 48 to 72 hours after egg laying (AEL), at which point the larvae were shifted to 29°C during 72 hours to induce *UAS-rpr*. The discs were examined immediately after the end of the shift to 29°C.

**Analysis of glial cells response during the recovery period.** *GMR-Gal4 tub-Gal80$^{ts}$ UAS-rpr* larvae were raised at 17°C until 144 ± 12 hours AEL and then shifted to the restrictive temperature (29°C) during 24 hours to overexpress *UAS-rpr* and then shifted back to the permissive temperature to allow them to recover. Discs were analysed immediately after the end of the shift to 29°C (T0), after 24 hours of recovery at permissive temperature (T1), and after 48 hours of recovery (T2).

The analysis of the effects caused by the ectopic expression in the retinal region upon apoptotic induction or in undamaged discs of the different *UAS* lines used in our analysis was performed by crossing *UAS-rpr*; *GMR-Gal4 tub-Gal80$^{ts}$ /CyO*, and *GMR-Gal4 tub-Gal80$^{ts}$ /CyO* with the following stocks:

*UAS-GFP*, *UAS-bsk$^{DN}$*, *UAS-puc2B*, *UAS-dpp$^{RNAi2}$*, *UAS-dpp$^{RNAi2}$; UAS-hh$^{RNAi}$*
*UAS-dpp$^{RNAiInt}$*, *UAS-dpp$^{RNAi33}$*, *UAS-hh$^{RNAi}$*, *UAS-dpp*, *UAS-hh-GFP*, *UAS-hh;UAS-dpp*, *puc-lacZ*, *dpp-lacZ*, *hh-lacZ*.

We also crossed *UAS-rpr*: *If /CyO; UAS-hh$^{RNAi}$/TM6B* to the following:
*GMR-Gal4 tub-Gal80$^{ts}$ /CyO; UAS-dpp$^{RNAiInt}$/TM6B*
*GMR-Gal4 tub-Gal80$^{ts}$ /CyO; UAS-dpp$^{RNAi33}$/TM6B*.

Larvae were raised at permissive temperature (25°C) until 48 to 72 hours AEL, at which point the larvae were shifted to 29°C during 72 hours for inducing *UAS-rpr*. The discs were examined immediately after the end of the shift to 29°C.

To study the effects caused by the ectopic expression in glial cells of the different *UAS* lines used in our analysis, we cross *tub-Gal80$^{ts}$/Cyo; repo-Gal4/TM6B* by the following stocks:

*UAS-hep$^{CA}$ (II) (III)*, *UAS-tkv$^{QD}$*, *UAS-brk*, *UAS-bsk$^{DN}$ UAS-ptc*, *UAS-dad*, *UAS-ci$^{RNAi}$*, *UAS-ci$^{(m1-3*103)}$*, *UAS-rbf$^{280}$*, *UAS-ptc;UAS-brk*, and *UAS-dad; UAS-ci$^{RNAi}$*.

Larvae were raised at 17°C until 48 to 72 hours AEL, at which point the larvae were shifted to 29°C for 72 hours. The discs were examined immediately after the end of the shift to 29°C.

We also crossed *repo-Gal4 UAS-ihog/TM6B* by *UAS-tkv$^{QD}$, and UAS-ci$^{(m1-3^* 103)}$*. In these experiments, larvae were raised at 25˚C, and the discs were dissected at 110 to 130 hours AEL.

**Analysis of glial cells response in leg discs.**    To analyse the effects on glial cells behaviour after inducing apoptosis in the leg region, *UAS-rpr*; *Dll-Gal4 tub-Gal80$^{ts}$/+*, or *UAS-rpr*; *hh-Gal4 tub-Gal80$^{ts}$/+* larvae were raised at 17˚C until 120 ± 12 hours AEL and then shifted to the restrictive temperature (29˚C) during 48 hours for inducing *UAS-rpr*. The discs were examined immediately after the end of the shift to 29˚C.

The analysis of the effects caused by the ectopic expression in the leg discs upon apoptotic induction or in undamaged discs of the different *UAS* lines used in our analysis was performed by crossing *UAS-rpr*; *Dll-Gal4 tub-Gal80$^{ts}$/+* and *Dll-Gal4 tub-Gal80$^{ts}$ /CyO* to the following stocks:

*UAS-mCD8-GFP*, *UAS-dpp$^{RNAi33}$*, *UAS-hh$^{RNAi}$*, *UAS-dpp*, *UAS-hh-GFP*, *UAS-hh;UAS-dpp*, *dpp-lacZ*, *dad-lacZ*, *BAC-Hh*:*GFP and TREGFP*.

We also crossed the following:

*UAS-rpr*; *Dll-Gal4 tub-Gal80$^{ts}$ CyO*; *UAS-hh$^{RNAi}$/TM6B* with *UAS-dpp$^{RNAi33}$*/TM6B
*UAS-rpr/ Cyo*; *hh-Gal4 tub-Gal80$^{ts}$* with *hep$^{r75}$/FM7GFP*
*dpp$^{d12}$/ Cyo ActGFP*; *hh-Gal4 tub-Gal80$^{ts}$ /TM6* with *UAS-rpr*; *dpp$^{d12}$ /Cyo*.

Larvae were raised at permissive temperature (17˚C) until 120 ± 12 hours AEL and then shifted to the restrictive temperature (29˚C) during 48 hours. The discs were examined immediately after the end of the shift to 29˚C.

**Combination of QF/QUAS systems.**    We crossed *GMR-QF*; *tub-Gal80$^{ts}$/Cyo* flies by the following: *UAS-bsk$^{DN}$*; *QUAS-rpr repo-Gal4, UAS-dad*; *QUAS-rpr repo-Gal4* and *UAS-ptc*; *QUAS-rpr repo-Gal4*, or *GMR-QF*; *tub-Gal80$^{ts}$;TM6/UAS-brk* by *QUAS-rpr repo-Gal4, UAS-dad*; *QUAS-rpr repo-Gal4* and *UAS-ptc*; *QUAS-rpr repo-Gal4*.

We also crossed *GMR-QF*; *tub-Gal80$^{ts}$/UAS-mCherry* with *BAC-Hh*:*GFP*; *QUAS-rpr repo-Gal4*.

To block *JNK*, dpp, and hh signalling in WG cells, we crossed *GMR-QF*; *Mz97-Gal4 UAS-GFP/Cyo* with the following stocks:

*UAS-ptc*; *QUAS-rpr*, *UAS-bsk$^{DN}$*; *QUAS-rpr*, *UAS-puc*; *QUAS-rpr*.
We also crossed *GMR-QF*; *UAS-brk/TM6* with *Mz97-Gal4 UAS-GFP*; *QUAS-rpr*.

## Immunocytochemistry

Immunostaining of the wing discs was performed according to standard protocols. The following primary antibodies were used: rabbit anti-Phospho-histone 3 1:200 (Cell Signaling Technology, Danvers, Massachusetts, USA), rabbit anti-cleaved Dcp1 1:200 (Cell Signaling Technology, Danvers, Massachusetts, USA), mouse anti-Patched 1:500 (Gift from Isabel Guerrero), mouse anti-ß Galactosidase 1:200 (Promega, Madison, Wisconsin, USA), anti-ß Galactosidase 1:500 (Cappel, MP Biomedicals, Santa Ana, California, USA), rat anti-Phospho-Mad 1:100 (from Ginés Morata), and Rabbit Anti-Dpp (Gift from Matthew Gibson) [45]. Mouse anti-Repo and Rat anti-Elav 1:200 were obtained from the Developmental Studies Hybridoma Bank at the University of Iowa. Secondary antibodies (Molecular Probes, Oregon, USA) were used at dilutions of 1:200.

Imaginal discs were mounted in Vectashield mounting fluorescent medium (Vector Laboratories, Burlingame, California, USA).

## EDU staining

Eye imaginal discs were dissected in PBS and then incubated in a solution with EdU for 1 hour at room temperature to label cells in S phase. Alexa Fluor detection was performed according to Click-iT EdU Fluor Imaging Kit (Invitrogen, Waltham, Massachusetts, USA).

## Generation of *puc cis*-regulatory modules reporters

We selected 2 different regions of the regulatory region of the gene *puckered* (puc-1 and puc-2; see S7 Fig) on the basis of the published open chromatin profile of eye discs during tumour development [32]. The selected region were amplified using KOD enzyme (Novagen, Gibbstown, New Jersey, USA) by PCR using the following primers:

PUC1: Forward: CAGTAAGCTTGCCGTCAACTTTTATCTGCCAACG
Reverse: CAGTAGATCTCGGGCTAATTGGACTGGGGTTCAA
PUC2: Forward: CAGTAAGCTTGGGGTGGCAATGACTCACAATAGG
Reverse: CAGTAGATCTCTGCAAAGATACATGCGGATCGG

The PCR products were cloned into the attB-hs44-nuc-lacZ vector using the HindIII and BglII restriction endonucleases (NEB, Ipswich, Massachusetts, USA).

The 2 sequences (*puc1* and *puc2*) were further subdivided into smaller overlapping fragments. The *puc1* region of 2782 bp was divided into 3 fragments: *puc1A* (978 bp), *puc1B* (938 bp), and *puc1C* (839 bp), while the *puc2* region of 1,636 bp was subdivided into 2 fragments: *puc2A* (897 bp) and *puc2B* (578 bp). To this end, PCR were performed using the aforementioned attB-hs44- puc1-nuc-lacZ and attB-hs44-puc2nuc-lacZ constructs as template. We added targets for the restriction enzymes BglII and HindIII to all the fragments generated and then cloned into the attB-hs44-nuc-lacZ vector using these restriction sites.

## Quantitative analysis

Images were processed using ImageJ software (NUH, Bethesda, USA).

We calculated glial density as the ratio between the number of glial cells, detected by the expression of Repo and the size of the region occupied by glial cells behind the MF in $\mu m^2$, without including the optical nerve (Repo positive cells/size of the area in $\mu m^2$). In Fig 1B', for instance, this region would correspond to the region expressing GFP (green), without including the region of the optical stalk (marked with a blue dotted line). Eye discs were measured using ImageJ.

Glial cell division was calculated as the ratio between the number of PH3-positive glial cells and the size of the region posterior to the MF in $\mu m^2$ (PH3-positive glial cells/size of the area in $\mu m^2$) or as the percentage of PH3-positive glial cells (PH3-positive glial cells /Total glial cells *100).

The percentage of glial cells incorporating EdU was calculated dividing the number of glial cells incorporating EdU by the total number of glial cells.

We calculated the density of WG cells, dividing the number of glial cells expressing *UAS-GFP* under the control of *Mz97-Gal4* by the area of the region occupied by glial cells in $\mu m^2$.

The proportion of WG cells in each eye discs analysed was calculated as the ratio between the number of glial cells expressing *UAS-GFP* under the control of *Mz97-Gal4* and the total number of glial cells (*Mz97-Gal4 UAS-GFP* glial cells/Repo positive cells glial).

Nuclei size was automatically calculated using the ImageJ software (NUH). We adjusted threshold intensity to visualise the nuclei of WG cells expressing *UAS-GFP* under control of *Mz97-Gal4*. Then, we automatically calculated the area size of each nucleus using the Area option in Set Measurements. We exclude the areas corresponding to the fusion of 2 or more glial cells.

The percentage of glial cells expressing the reporter *puc2B* was calculated dividing the number of glial cells that expressed *puc2B-lacZ* at detectable levels by the total number of glial cells in each eye.

To calculate the number of glial cells that expressed *puc2B-lacz* at high levels in each eye disc analysed, we used ImageJ to automatically calculate the intensity of expression of *puc2B-lacZ* in all glial cells of the discs. Then we defined the percentage of cells that express the reporter at high levels; we established a threshold where the maximum value corresponds to the maximum value for that specific sample, and the minimum value corresponds to a value 10% less than the maximum value. In this sense, we have an internal control for each sample to compare between different images.

To calculate the motility of glia cells, we selected several points of the glial front at the start of the movie and the distance to the closest perpendicular point at the end of the movie was measured. If the end point is closer to the MF than the start point, then the distance covered is considered positive. If opposite, the distance is considered negative ($p$ value = 0.0172, $n$ = at least 3 independent movies).

Images were adjusted for display using ImageJ (Fiji, NUH, Bethesda, USA). Profile plots were made using "plot profile" in Fiji, based on the average intensity of the image across the x-axis of the region indicated in the figures. Discs were stained and imaged alongside sibling controls, ensuring identical conditions for each group.

## In vitro culture

Imaginal discs were cultured as described [74].

## Statistical analysis

For statistical tests applied to each experiment, n- and $p$-values, please see tables in S1 and S2 Text. p-Values shown on the graphs are indicated with the following asterisk codes: $^*p < 0.5$; $^{**}p < 0.01$; $^{***}p < 0.001$; $^{****}p < 0.0001$.

The error bars indicate the standard error of the mean (SEM).

## Microscopy

Images were captured using a Confocal LSM510 Vertical Zeiss (Oberkochen, Germany) and processed with ImageJ or Adobe Photoshop CS4 (San Jose, California, USA).

## Supporting information

**S1 Fig. *UAS-rpr* expression under the control of *GMR-Gal4* induces apoptosis in the eye imaginal disc.** GMR, glass multiple reporter.
(TIF)

**S2 Fig. Glia proliferation after inducing cell death in the retinal region discs.**
(TIF)

**S3 Fig. Pattern of proliferation of glial cells at different times after inducing apoptosis.**
(TIF)

**S4 Fig. WG cells generate new projections in direction to the damage area.** WG, wrapping glia.
(TIF)

**S5 Fig. Apoptotic induction in retinal cells promotes phagocytic activity in glial cells.**
(TIF)

**S6 Fig. JNK pathway is ectopically activated in glial cells in response to damage in the retina.** JNK, c-Jun N-terminal kinase.
(TIF)

**S7 Fig. JNK signalling is activated in response to damage.** JNK, c-Jun N-terminal kinase.
(TIF)

**S8 Fig. *puc* reporter.** *puc*, *puckered*.
(TIF)

**S9 Fig. The co-overexpression of *UAS-bsk*$^{DN}$ and *UAS-rpr* under the control of *GMR-Gal4* induces apoptosis in the eye imaginal disc.** GMR, glass multiple reporter.
(TIF)

**S10 Fig. *eiger* is not required during the development of glial cells in the eye disc.**
(TIF)

**S11 Fig. Glial motility is affected in *hep*$^{r75}$ mutant discs.**
(TIF)

**S12 Fig. The overexpression of *eiger* in the retina region is not sufficient for activating JNK signalling in glial cells.** JNK, c-Jun N-terminal kinase.
(TIF)

**S13 Fig. Overexpression of *hep*$^{CA}$ does not increase the number of glial cells in the eye discs.**
(TIF)

**S14 Fig. Expression of *dpp-LacZ* and pMad increase in damaged discs.**
(TIF)

**S15 Fig. Expression of Hh and ptc in damaged discs.** Hh, Hedgehog; ptc, patched.
(TIF)

**S16 Fig. Hh localises in the apical area of the eye discs.** Hh, Hedgehog.
(TIF)

**S17 Fig. The down-regulation of Dpp and Hh signalling reduces glial cells response.** Dpp, Decapentaplegic; Hh, Hedgehog.
(TIF)

**S18 Fig. The overexpression of *dpp* and *hh* induces over migration and proliferation of glial cells.** Dpp, Decapentaplegic; Hh, Hedgehog.
(TIF)

**S19 Fig. Ptc is expressed at high levels in discs overexpressing *ihog*.** ptc, patched.
(TIF)

**S20 Fig. Overexpression of *dpp* in the retina region induces activation of JNK signalling in glial cells.** Dpp, Decapentaplegic; JNK, c-Jun N-terminal kinase.
(TIF)

**S21 Fig. The down-regulation of JNK signalling does not impair the ability of glial cells to engulf cellular debris.** JNK, c-Jun N-terminal kinase.
(TIF)

**S22 Fig. JNK signalling is mediating the glial response triggers after cell death induction in the leg discs.** JNK, c-Jun N-terminal kinase.
(TIF)

**S23 Fig. Hh signalling is not activated in glial cells in the leg discs in response to apoptotic induction.** Hh, Hedgehog.
(TIF)

**S24 Fig. Dpp signalling is activated after cell death induction in the leg discs, and its function is necessary for inducing glial response.** Dpp, Decapentaplegic.
(TIF)

**S25 Fig. The down-regulation of Dpp signalling reduces glial migration in the leg discs.** Dpp, Decapentaplegic.
(TIF)

**S26 Fig. The up-regulation of Dpp and Hh signalling increases glial migration in the leg.** Dpp, Decapentaplegic; Hh, Hedgehog.
(TIF)

**S1 Text. Statistical analysis tables of main figures.**
(PDF)

**S2 Text. Statistical analysis tables of Supporting information figures.**
(PDF)

**S3 Text. Figure legends of Supporting information figures.**
(DOCX)

**S1 Data. Raw numerical values of main figures.**
(XLSX)

**S2 Data. Raw numerical values of Supporting information figures.**
(XLSX)

**S1 Video. Confocal time-lapse imaging of damaged *GMR-QF; UAS-GFP Mz97-Gal4; QUAS-rpr* eye disc.** GMR, glass multiple reporter.
(AVI)

**S2 Video. Confocal time-lapse imaging of control *UAS-GFP Mz97-Gal4* eye discs.**
(AVI)

**S3 Video. Time-lapse imaging showing the motility of glial cells in control eye discs.** Glial cells are labelled with *UAS-GFP* using *repo-Gal4*.
(AVI)

**S4 Video. Time-lapse imaging showing the motility of glial cells in *hep^{r75}*; *repo-Gal4 UAS-GFP* mutant eye discs.**
(AVI)

**S5 Video. Confocal time-lapse imaging of damaged *GMR-QF/UAS-bsk^{DN}; UAS-GFP Mz97-Gal4; QUAS-rpr* eye disc.** *Mz97-Gal4* drives *UAS-bsk^{DN}* and *UAS-GFP* expressions to reveal WG cells. GMR, glass multiple reporter; WG, wrapping glia.
(AVI)

## Acknowledgments

We thank Jose Felix de Celis and Luis Alberto Baena for providing reagents and useful discussion and Claire Hills for helping to improve the manuscript. We are very grateful to Gines Morata, Isabel Guerrero, Hermann Steller, Sergio Casas, Matthew C. Gibson, the Bloomington

Stock Center, and the Developmental Studies Hybridoma Bank for providing fly strains and antibodies.

## Author Contributions

**Conceptualization:** Antonio Baonza.

**Data curation:** Antonio Baonza.

**Formal analysis:** Sergio B. Velarde, Alvaro Quevedo, Antonio Baonza.

**Funding acquisition:** Antonio Baonza.

**Investigation:** Sergio B. Velarde, Alvaro Quevedo, Carlos Estella, Antonio Baonza.

**Methodology:** Sergio B. Velarde, Alvaro Quevedo.

**Supervision:** Antonio Baonza.

**Writing – original draft:** Antonio Baonza.

**Writing – review & editing:** Sergio B. Velarde.

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
