## [Editor Report · Decision Letter 0]

23 Apr 2021

Dear Antonio, 

Thank you for submitting the revision of your manuscript entitled "Dpp and Hedgehog promote the Glial response to neuronal apoptosis in the developing Drosophila Visual system" for consideration as a Research Article by PLOS Biology.

Your revision has now been evaluated by the PLOS Biology editorial staff as well as by the original academic editor and I am writing to let you know that we would like to send it out for external peer review.

Because this is a new submission, before we can send your manuscript to the reviewers, we need you to complete your submission by providing again the metadata that is required for full assessment. To this end, please login to Editorial Manager where you will find the paper in the 'Submissions Needing Revisions' folder on your homepage. Please click 'Revise Submission' from the Action Links and complete all additional questions in the submission questionnaire.

Please re-submit your manuscript within two working days, i.e. by Apr 27 2021 11:59PM.

Kind regards,

Ines

--

Ines Alvarez-Garcia, PhD,

Senior Editor

PLOS Biology

---

## [Decision Letter · Decision Letter 1]

27 May 2021

Dear Dr Baonza,

Thank you very much for submitting a revised version of your manuscript entitled "Dpp and Hedgehog promote the Glial response to neuronal apoptosis in the developing Drosophila Visual system" for consideration as a Research Article at PLOS Biology. This revised version of your manuscript has been evaluated by the PLOS Biology editors, the Academic Editor and three of the original reviewers.

You will see that the reviewers think the manuscript has notably improved and that the findings remain interesting, however they would like you to improve the general structure of the manuscript and to streamline the writing to make the text more focused. Reviewer 3 proposes a clone analysis to improve the resolution of the engulfment process, but we won’t make this point essential for resubmission. However, you should address all the other points.

In light of the reviews (attached below), we are pleased to offer you the opportunity to address the remaining points raised by the reviewers in a revised version that we anticipate should not take you very long. We will then assess your revised manuscript and your response to the reviewers' comments and we may consult the reviewers again.

We expect to receive your revised manuscript within 1 month.

**IMPORTANT - SUBMITTING YOUR REVISION**

3. Resubmission Checklist

a) *Published Peer Review*

b) *PLOS Data Policy*

Please provide the data underlying the following figures, and make sure you mention in the corresponding figure legends WHERE THE DATA CAN BE FOUND:

Fig. 1G, H; Fig. 2I-L, P; Fig. 3E, F; Fig. 4Q-S; Fig. 5I-K; Fig. 6H, I, N, O; Fig. 7O; Fig. 8G, H, O, P; Fig. 9G-I; Fig. 10K-M; Fig. 11C, D; Fig. S2E, F; Fig. S3E; Fig. S4E; Fig. S10D-F; Fig. S11; Fig. S13I-K; Fig. S17A-D; Fig. S18E, F; Fig. S20E and Fig. S22I

c) *Financial Disclosure*

Please include grant numbers and the URLs of any funder's website. Use the full name, not acronyms, of funding institutions, and use initials to identify authors who received the funding.

Describe the role of any sponsors or funders in the study design, data collection and analysis, decision to publish, or preparation of the manuscript. If the funders had no role in any of the above, include this sentence at the end of your statement: "The funders had no role in study design, data collection and analysis, decision to publish, or preparation of the manuscript."

Sincerely,

Ines

--

Ines Alvarez-Garcia, PhD

Senior Editor

PLOS Biology

Reviewers’ comments

Rev. 1:

This manuscript utilized the eye disc to examine how glia cells respond to the neuronal apoptosis. The authors performed the extensive amount of works and provided evidences for the roles of Hh and Dpp in regulating glial proliferation and motility after inducing neuronal apoptosis. Overall, this work provides detailed framework for glial responses upon damages in the eye disk. I have a few points that need to be considered before publication.

1. It is clear that glia changes their density and location through DPP/Hh/JNK mediated cell proliferation or migration in the damaged tissues. However, it is still unclear the physiological roles of glial response upon neuronal apoptosis in the eye disk. The authors showed that membrane protrusions of glial cells through DPP/JNK signaling contain neuronal debris. Do these mis-localized-glial cells play critical roles in eliminating neuronal debris? Is regeneration of photoreceptors or tissue maintenance impaired when glial responses are reduced? Addressing these question will provide further rationale and significance of the proposed work.

2. There are too many similar experiments in the current manuscript. For better and clear representation of the manuscript, I would suggest at least shortening the experimental details in Fig 11, which is about replicated data using the leg disk.

Rev. 2:

The manuscript by Velarde et al. has undergone a number of changes since the first submission, and the majority of my concerns have been addressed. Clearly a lot of work has gone into this paper, which also led to a lengthy revision that could be refined in some parts. In particular, the new addition of the leg disc data (figure 11) and some of the additional supplemental data feel a bit cumbersome in places. One thing that could help throughout the text is to include more references to the specific parts of the figures as the relevant information is discussed, and that all portions of the figures are referred to in the corresponding text. In many areas this was only partially done, or out of order, which led to additional confusion. However, my major concerns about the manuscript have been addressed.

Rev. 3:

In the rebuttal letter, the authors describe that they tightened the narrative of the manuscript. However, a study with 12 main figures, 26 supplemental figures and 30 pages of results, lacks in my view concision. Focus is essential to document the key points, evidence for which is difficult to find or may not exist.

In their rebuttal of reviewer comments, the authors now describe their experimental set-up as a response to cell death (and no longer as an injury model, which is more justified). A key conclusion of the study is that apoptotic cells in the eye imaginal disc release Hh and Dpp to influence glial responses. However, the data do not include clear images revealing that co-localisation with Dcp-1 and markers, indicative of Dpp or Hh production, indeed occurs.

Responses seem to be general and not local, influencing glial cell density. However, for instance in Figure 1 E and F, the density does not appear to be increased, only quantifications support the point. Also in Figure 2E, a prominent cluster of apoptotic cells is shown, but in the figure one cannot detect whether there are more glial cells affiliated with the cluster or not, indicative of a local density change.

Another key point is the engulfment of apoptotic debris and morphological changes of glia. The eye discs contain a high number of small Dcp-1 positive particles (whose nature is not clear, the antibody normally labels very strongly apoptotic nuclei, but it is quite difficult to evaluate dotty particles, which can occur also in areas without cell death). The images are shown in low magnification making it very difficult to see, whether this is true engulfment or to pinpoint which morphological changes occured. I am thus wondering whether the analysis of clones would help to assess behaviors in high resolution and to untangle general eye-disc responses from the specific glial response.

Again in my view, it is paramount to shorten the manuscript substantially to highlight the key evidence for each of the main points presented in the model. These would need to be supported by high resolution images.

Minor comments :

In the model, it is proposed that there is increased motility in the photoreceptor epithelium, but it is unlikely that photoreceptors migrate ?

Information about statistical tests is still only provided in a Supplemental Table and not in the main text.

---

## [Editor Report · Decision Letter 2]

16 Jul 2021

Dear Dr Baonza,

On behalf of my colleagues and the Academic Editor, Richard Daneman, I am pleased to say that we can in principle offer to publish your Research Article entitled "Dpp and Hedgehog promote the glial response to neuronal apoptosis in the developing Drosophila visual system" in PLOS Biology, provided you address any remaining formatting and reporting issues. These will be detailed in an email that will follow this letter and that you will usually receive within 2-3 business days, during which time no action is required from you. Please note that we will not be able to formally accept your manuscript and schedule it for publication until you have made the required changes.

PRESS

Sincerely, 

Ines

--

Ines Alvarez-Garcia, PhD 

Senior Editor 

PLOS Biology
